# A computational framework for modelling infectious disease policy based on age and household structure with applications to the COVID-19 pandemic

Joe Hilton[1,2]*, Heather Riley[3], Lorenzo Pellis[3,4], Rabia Aziza[1,2], Samuel P. C. Brand[1,2,5], Ivy K. Kombe[5], John Ojal[5,6], Andrea Parisi[1,2], Matt J. Keeling[1,2,7], D. James Nokes[1,2,5], Robert Manson-Sawko[8], Thomas House[3,4,8]

**1** School of Life Sciences, University of Warwick, Coventry, United Kingdom, **2** Zeeman Institue (SBIDER), University of Warwick, Coventry, United Kingdom, **3** Department of Mathematics, University of Manchester, Manchester, United Kingdom, **4** The Alan Turing Institute for Data Science and Artificial Intelligence, London, United Kingdom, **5** Kenya Medical Research Institute - Wellcome Trust Research Programme, Kilifi, Kenya, **6** Department of Infectious Disease Epidemiology, London School of Hygiene & Tropical Medicine, London, United Kingdom, **7** Mathematics Institute, University of Warwick, Coventry, United Kingdom, **8** IBM Research Europe, Hartree Centre, Daresbury, United Kingdom

* joe.b.hilton@gmail.com

**Data Availability Statement:** Our main model code is openly available at https://github.com/JBHilton/hh-npi-modelling. The code used to generate the

## Abstract

The widespread, and in many countries unprecedented, use of non-pharmaceutical interventions (NPIs) during the COVID-19 pandemic has highlighted the need for mathematical models which can estimate the impact of these measures while accounting for the highly heterogeneous risk profile of COVID-19. Models accounting either for age structure or the household structure necessary to explicitly model many NPIs are commonly used in infectious disease modelling, but models incorporating both levels of structure present substantial computational and mathematical challenges due to their high dimensionality. Here we present a modelling framework for the spread of an epidemic that includes explicit representation of age structure and household structure. Our model is formulated in terms of tractable systems of ordinary differential equations for which we provide an open-source Python implementation. Such tractability leads to significant benefits for model calibration, exhaustive evaluation of possible parameter values, and interpretability of results. We demonstrate the flexibility of our model through four policy case studies, where we quantify the likely benefits of the following measures which were either considered or implemented in the UK during the current COVID-19 pandemic: control of within- and between-household mixing through NPIs; formation of support bubbles during lockdown periods; out-of-household isolation (OOHI); and temporary relaxation of NPIs during holiday periods. Our ordinary differential equation formulation and associated analysis demonstrate that multiple dimensions of risk stratification and social structure can be incorporated into infectious disease models without sacrificing mathematical tractability. This model and its software implementation expand the range of tools available to infectious disease policy analysts.

UK household composition distribution is openly available at https://github.com/JBHilton/processing-household-composition-data.

**Funding:** This work was supported by the UK Foreign, Commonwealth and Development Office (FCDO) and Wellcome Trust-funded CIMEA grant (220985/Z/20/Z), and National Institute for Health and Care Research (NIHR) Global Health Research Project GeMVi (17/63/82), grant recipient DJN, as well as the Hartree National Centre for Digital Innovation, a collaboration between STFC and IBM, grant recipient RMS. TH is supported by the Royal Society (grant number INF/R2/180067). TH and LP are supported by the UK Research and Innovation-funded JUNIPER consortium (grant number MR/V038613/1) and the Alan Turing Institute for Data Science and Artificial Intelligence. TH, LP and HR are supported by the UK Research and Innovation COVID-19 rolling scheme (grant number EP/V027468/1). RMS received a salary from IBM Research Europe. The funders had no role in study design, data collection and analysis, decision to publish, or preparation of the manuscript.

**Competing interests:** The authors have declared that no competing interests exist.

## Author summary

Non-pharmaceutical interventions have seen widespread use during the COVID-19 pandemic. Some of the most prominent such interventions act at the household level, with isolation measures confining individuals to their own home and measures such as work and school closure seeking to prevent transmission between members of different households. In this study we develop a mathematical model of COVID-19 transmission in a population of households which accounts for age-specific variation in behaviour. We demonstrate how to perform simulations with our model as well as how to calibrate it to empirical estimates of epidemic growth rate. We apply our model to four specific policy questions, including the impact of within-household controls on transmission, the effect of support bubble exemptions to between-household mixing controls, the effect of temporary relaxations to non-pharmaceutical interventions, and the possible impact of out-of-household isolation measures. We provide an open source software implementation of our model so that it can be used by researchers and policy experts interested in planning interventions during subsequent stages of the COVID-19 pandemic and other future pandemics.

This is a *PLOS Computational Biology* Methods paper.

## Introduction

The current COVID-19 pandemic, caused by the SARS-CoV-2 coronavirus first detected in the human population in late 2019, has at the time of writing led to over 6.3 million confirmed deaths arising from over 530 million confirmed cases [1], with the number of infections likely to be much higher than confirmed cases due to factors such as asymptomatic infection [2]. While over 11 billion vaccine doses have now been distributed [1], in the earlier phases of the pandemic governments throughout the world mounted various non-pharmaceutical intervention (NPI) intended to reduce transmission [3].

Some NPIs have included measures common to many infectious disease outbreaks such as hand and surface hygiene, some have included extension of measures commonly used for infection control in clinical contexts (such wearing facemasks) to the community, and others such as social distancing were more novel [4]. The possibility that masking and hand hygiene could reduce spread of seasonal influenza, including within households, was considered before the COVID-19 pandemic [5, 6], and also subject to study during it [7].

While these NPIs sought to reduce transmission in all contexts, others were primarily targeted towards reducing transmission between households, including (TTI) policies that led to detected cases isolating at home or the more general "lockdown" measures in which significant restrictions were placed on between-household mixing for the whole population [3, 8].

Clearly, quantifying the likely impact of NPIs on infectious disease transmisison is important in formulation of an appropriate policy response, and the inability to perform fully controlled experiments in a pandemic situation means that mathematical modelling is often a key tool [9]. In this work, we present a contribution to the attempt to model the impact of NPIs with particular attention to the role of household structure. Households are the most fundamental unit of demographic aggregation and cause individuals to engage in repeated close contacts that are likely to spread infection, and as such have been an important part of epidemic modelling since the earliest days of the discipline [10].

As such, household analyses have been carried out as part of the response to almost all significant outbreaks, and in the context of the COVID-19 pandemic numerous studies have considered household transmission [11]. These have tended, however, to be based on final outcomes in households, which is not appropriate for a situation that is rapidly changing over time [12]. Where time is explicitly included in models with household structure, this is often achieved through individual-based stochastic simulation, and can generate important insights to questions such as the expected impacts of different (TTI) policies [13].

Household structure alone is, however, likely to be insufficient to capture the impacts of NPIs. Combination of both age and household structure in epidemic models is sometimes necessary even for prediction of simple epidemiological quantities such as peak prevalence of infection or final outbreak size in the absence of interventions [14]. When attempting to predict more complex outcomes, and particularly the impact of interventions that involve households in a situation of significant heterogeneity in risk by age, it is however essential to include both in models. In particular, there is significant additional complexity generated by the process of transmission, with a mosaic of mixing patterns between people of different ages known to have a significant impact on infectious disease dynamics [15]. Since the highly influential POLYMOD study of Mossong *et al.* [16] these mixing patterns have increasingly been the subject of direct empirical observation, and this continued through the COVID-19 pandemic with studies such as CoMix [17]. As well as the impacts of age on mixing, there has been consistent strong evidence that age plays an important role in determining risk from COVID-19 [18], with orders of magnitude of variability in risk of death due to COVID-19, with infection fatality ratios (IFRs) estimated from under 0.001% in the 5–9 age group to over 10% in the 80+ age group [19]. Together, variable age IFRs and mixing patterns are believed to be responsible for significant variability in outcomes between different countries [19, 20]. Evidence for the impact of age on transmissibility and susceptibility has been more mixed [21–24].

In this paper we introduce an age- and household-structured model which expands on previous approaches by incorporating detailed data-driven age-stratified household composition structures while retaining a mathematically tractable formulation in terms of ordinary differential equations. Analysis of such a differential-equation approach was provided for independent households by Bailey [25], with the 'self-consistent' equations needed to capture between-household transmission given by Ball [26]. Analytical and numerical approaches to these self-consistent equations were provided by House *et al.* [27] and Ross *et al.* [28], demonstrating that these equations can have significant benefits over individual-based stochastic simulations, particularly when a comprehensive 'sweep' over parameters or exact calculation of quantities such as early exponential doubling times is required. Hilton and Keeling [29] showed that this methodology could be applied to the extremely large dimensions present in demographically realistic populations. This study focused exclusively on early invasion dynamics and stochastic equilibria of demographically structured epidemic models, with the modelling structure it introduced unable to directly simulate the transient dynamics of an epidemic and thus unable to model many standard control policies which act "in real time". One of the aims of the present study is to expand on this and previous approaches with a model which accounts for demographic structure while also allowing for direct simulation of transient epidemiological dynamics.

In the Methods section of this paper, we introduce a formal mathematical structure for this age- and household-structured model. We derive a formula for the susceptible-infectious transmission probability (SITP), the probability that a susceptible individual is infected by an index case in their own household, in terms of the within-household transmission rate, allowing us to calibrate this within-household transmission rate to estimates of SITP which have been calculated from observational studies [30]. We also derive an Euler-Lotka equation for

our model, from which we can estimate a between-household transmission rate from empirical estimates of the growth rate of cases. This estimation method allows us to calibrate our model to the population-level state of an epidemic at specific points in its evolution. This makes our model ideally suited to a workflow taking a growth rate or doubling time estimate as an input and returning projections of the impact of specific NPIs as an output. These two formulae allow us to calibrate both the within- and between-household transmission intensities in our model to specific phases in the evolution of an epidemic based on empirical observations of the spread of infection. We then show how epidemic trajectories can be estimated by solving a system of ODEs. We use this approach to perform interpretable parameter sweeps over the decision space. The large dimensionality of the differential equation system presents a computational challenge, and we provide an open-source implementation in Python of our methods, which can be applied to quite general epidemiological scenarios. While vaccination is beyond the scope of our current work, we also note that analytic results for critical vaccination thresholds should be available for the model we describe [31, 32].

A wealth of modelling studies of non-pharmaceutical interventions have been published since the beginning of the pandemic, using frameworks including stochastic and deterministic compartmental models, branching processes, network models, and agent-based microsimulation models, covering policies ranging from contact tracing to travel restrictions to school closures, as well as possible exit strategies when interventions are withdrawn [33–40]. In the present study we introduce an approach which can model the interplay between household structure and control policy in a deterministic compartmental setting; while microsimulation models of COVID-19 have been developed which can account for household structure, they must be implemented through repeat simulation [41]. Although one study has considered a similar formalism to ours based on self-consistent equations in the context of NPIs during the COVID-19, it did not attempt to combine household structure with stratification based on age or other risk classes [9].

To demonstrate the wide range of possible applications of our model, we present four policy analyses based on commissions originally carried out for the UK Government's Scientific Pandemic Influenza Group on Modelling (SPI-M) committee, and reflected in documents such as the paper *Reducing within- and between-household transmission in light of new variant SARS--CoV-2, 14 January* from The Scientific Advisory Group for Emergencies [42]. We first demonstrate our model's basic capacity to distinguish between two levels of social contacts by comparing interventions targeted at the within-household level to those targeted at the between-household level. We then carry out an analysis of "support bubble" exemptions during lockdown periods which explores the impact of underlying household structure on the infectious disease dynamics predicted by our model. In a similar vein, we carry out an analysis of short-term relaxations of lockdown measures, during which members of distinct households are allowed to mix in specific limited capacities. This allows us to project the impact of allowances for mixing during festive periods taking place during lockdown periods, while also adapting our model to settings with a dynamically changing household contact structure. Our final policy analysis explores out-of-household isolation (OOHI), an alternative to within-household isolation which allows infectious individuals to isolate outside of their home in order to minimise transmission to clinically vulnerable members of their own household. Whereas the analyses of support bubbles and relaxations of lockdown measures focus on changing the underlying contact structure of the model by merging households together, in the analysis of OOHI we model the impact of changes to household contact structures caused by individuals leaving and rejoining a household. Between them, these examples demonstrate our models capacity to simulate not only the role of household-structured transmission in infectious disease dynamics, but also the impact of changes to household structure on these dynamics.

## Methods

### General modelling framework

Model parameters are provided for reference in Table 1, along with the values used in this work. In the first section of S1 Appendix we provide a description of the computational implementation of our model, as well as some basic benchmarking statistics.

Our model considers a population composed of a large number of individuals, with each individual belonging to exactly one household. Each individual is also a member of a risk class, which may correspond to an age band but could reflect other more general stratifications such as vulnerability to infection due to comorbidity or behavioural risk factors relating to infection such as non-socially distanced key work outside the home. Denoting these classes by $C_1, \ldots, C_K$, where $K$ is the total number of classes in our stratification, we define the *composition* of a household to be the vector $\mathbf{N} = (N_1, \ldots, N_K)$, where $N_i$ is the number of individuals of class $C_i$ who belong to the household. We will assume that the composition of a household is fixed, but we emphasise that "belonging" to a household may not reflect a permanent physical presence; our infectious dynamics assume individuals from different households mix at locations such as schools and workplaces, while our modelling of out-of-household isolation allows household members to temporarily leave the household and return after a period of isolation. Our assumption of a fixed household composition is based on the comparatively short time scales involved in our modelling, over which the population-level impact of births, deaths, or permanent movements between households on household compositions will be relatively small. Each household member also belongs to one of $L$ epidemiological compartments, which we denote by $X_1, \ldots, X_L$. The *state* of a household is given by the $(K \times L)$-dimensional vector $\mathbf{x} = (x_1^1, \ldots, x_1^L, \ldots x_K^1, \ldots x_K^L)$, with $x_i^j$ denoting the number of individuals in class $C_i$ who are

**Table 1. Parameter notation, choices, and sources for the age- and household structured infection model.**

| Model | Parameter | Notation | Value | Source |
|---|---|---|---|---|
| **All** | Doubling time | $T_2$ | Varies by case | |
| | Growth rate | $r$ | Varies by case | |
| | Susceptible-infectious transmission probability | $p_N$ | See Table 2 | House *et al.* [30] |
| | Density exponent | $d$ | Varies by case | Derived from SITP |
| | Within-household transmission rate | $\beta_{\text{int}}$ | Varies by case | Derived from SITP |
| | Between-household transmission rate | $\beta_{\text{ext}}$ | Varies by case | Derived from growth rate |
| | Within-household contact matrix | $K^{\text{int}}$ | | Prem *et al.* [43] |
| | Between-household contact matrix | $K^{\text{ext}}$ | | Prem *et al.* [43] |
| **SEIR** | Incubation rate | $\alpha$ | 1/1.16 | Hart *et al.* [44] |
| | Recovery rate | $\gamma$ | 1/9.64 | Hart *et al.* [44] |
| **SEPIR** | Prodromal relative transmission strength | $\tau_P$ | 3 | Hart *et al.* [44] |
| | Infectious onset rate | $\alpha_1$ | 1/1.16 | Hart *et al.* [44] |
| | Symptom onset rate | $\alpha_2$ | 1/4.64 | Hart *et al.* [44] |
| | Recovery rate | $\gamma$ | 1/5 | Hart *et al.* [44] |
| **SEPIRQ** | Prodromal relative transmission strength | $\tau_P$ | 3 | Hart *et al.* [44] |
| | Infectious onset rate | $\alpha_1$ | 1/1.16 | Hart *et al.* [44] |
| | Symptom onset rate | $\alpha_2$ | 1/4.64 | Hart *et al.* [44] |
| | Recovery rate | $\gamma$ | 1/5 | Hart *et al.* [44] |
| | Detection rate | $\delta$ | Varies in study | |
| | Isolation probability | $p_Q$ | Varies in study | |
| | Discharge rate | $\rho$ | 1/14 | Chosen for study |

in compartment $X_j$ such that $\sum_{j=1}^{L} x_i^j = N_i$. We emphasise that the composition refers only to the set of risk classes represented in a household and their frequencies, whereas the state captures the combination of risk classes and epidemiological compartments. This general formulation allows us to talk about different epidemiological and population stratifications using a single modelling framework. As an example, we can consider a simple two-age-class SIR model where the population is divided into adults and children (see, for example, Keeling and Rohani [45]). The $K = 2$ risk classes here are given by $C_C$ (children) and $C_A$ (adults), the $L = 3$ epidemiological compartments are $S$ (susceptible), $I$ (infectious), and $R$ (recovered) and the state of a household is given by $(S_C, I_C, R_C, S_A, I_A, R_A)$, where in keeping with convention we use upper-case letters for both the classes themselves and the frequency of these classes within the household.

The state of each individual household evolves stochastically according to a set of transition rates which summarise the epidemiological interactions and physiological processes relevant to infection. These rates define a transition matrix $\mathbf{Q}$, the $(i, j)$-th element of which, $Q_{i,j}$, gives the rate at which a household transitions from epidemiological state $i$ to epidemiological state $j$. In the limit of a large number of small households, it is possible to obtain a formal diffusion and deterministic limit to these stochastic dynamics [46]. In such a large population limit, the probability that a household chosen uniformly at random from the set of households is in a given state tends to the state distribution vector $\mathbf{H}$, which evolves according to the system of equations

$$\frac{d\mathbf{H}}{dt} = \mathbf{H}\mathbf{Q}. \tag{1}$$

The state distribution $\mathbf{H}$ can be interpreted as capturing the expected proportion of households in each epidemiological state in a population made up of a large number of these households. To model a population consisting of more than one type of household, we can construct a block-diagonal transition matrix whose blocks contain the transition events for each household composition. Because we assume that the composition of each household is fixed, there are no transitions from one household type to another and so there will be no nonzero elements except within these these diagonal blocks.

The transition matrix $\mathbf{Q}$ can be written as a sum of two components: $\mathbf{Q}_{\text{int}}$, which captures all the "internal" events in the household evolution, including infectious transmission from one member of the household to another and individual-level, physiological transitions between successive stages of infection and recovery, and $\mathbf{Q}_{\text{ext}}$, which captures external imports of infection into the household. We make this distinction because while the internal events take place at fixed rates which depend only on the current epidemiological state of the household, the external import events depend on the state distribution of the entire population of households. The population's evolution is then given by the nonlinear system of equations

$$\frac{d\mathbf{H}}{dt} = \mathbf{H}(\mathbf{Q}_{\text{int}} + \mathbf{Q}_{\text{ext}}(\mathbf{H}, t)). \tag{2}$$

Solving these equations gives us the proportion of households of each composition in each epidemiological state over time. From this we can calculate quantities like the expected number of class $C_i$ individuals per household who are in epidemiological compartment $X_j$, given by

$$\langle x_i^j(t) \rangle = \sum_k x_i^j(k) H_k(t), \tag{3}$$

where $x_i^j(k)$ is the number of class $C_i$ individuals in epidemiological compartment $X_j$ in state $\mathbf{x}_k = (x_1^1(k), \ldots, x_1^L(k), \ldots, x_K^1(k), \ldots, x_K^L(k))$. The mean number of class $C_i$ individuals per

household is given by

$$\langle N_i(t) \rangle = \sum_k \sum_j x_i^j H_k(t),$$ (4)

which is independent of time $t$ because household compositions are static. Dividing by average household size gives us the proportion of the population which belongs to class $C_i$ as $\langle N_i \rangle / \langle N \rangle$. In the SIR model with children and adults described above, for example, the prevalence of infection amongst children expressed as a proportion would be given by:

$$\langle I_C(t) \rangle = \frac{\sum_k I_C(k) H_k(t)}{\langle N_C \rangle}.$$

The proportion of households which are of composition $\mathbf{N} = (N_1, \ldots, N_K)$ is

$$h_{\mathbf{N}} = \sum_{k \in Z} H_k(t), \qquad Z = \left\{ k \mid \sum_j x_i^j(k) = N_i, \ \forall i \in [K] \right\}.$$

Here we write $[n]$ for the set of natural numbers from 1 to $n$ inclusive. Verbally, $Z$ is the set of states with the composition $\mathbf{N}$.

The proportion of class $C_i$ individuals which belong to households of composition $\mathbf{N}$ is then $h(\mathbf{N}|i) = N_i h_{\mathbf{N}} / \langle N_{C_i} \rangle$. This is identical to the probability that an arbitrary individual of class $C_i$ belongs to a household of composition $\mathbf{N}$, and so gives the probability that a newly infected individual of class $C_i$ in the early stages of an epidemic triggers a within-household infection in a household of composition $\mathbf{N}$.

## Modelling infection events

Our model allows for two basic routes of infection: within-household (or internal) infection, and between-household (or external) infection. Denote by $\mathcal{I}$ the set of indices of the epidemiological states which contribute to infectious pressure. In this study we will work with two compartmental structures, the susceptible-exposed-infectious-recovered (SEIR) and susceptible-exposed-prodromal-infectious-recovered (SEPIR) models. The two models differ by the absence and presence respectively of a prodromal class, containing individuals who are currently infectious but have not yet presented with symptoms. In the SEIR model, these prodromal individuals are still implicitly present in the model population but are modelled as being indistinguishable from the individuals in the (symptomatic) infectious class. In the SEIR model, the only state to contribute to infectious pressure is the infectious compartment ($\mathcal{I} = \{2\}$); in the SEPIR model, both the prodromal and symptomatic infectious compartments contribute to infection ($\mathcal{I} = \{2, 3\}$). In what follows, we assume that our compartmental structure includes a single susceptible compartment and that only individuals in this compartment can acquire infection, but our reasoning easily extends to models with multiple such compartments. We assume that the rate of transmission from an infectious individual to a susceptible one is directly proportional to the amount of time which those individuals spend in contact with one another.

**Within-household transmission.** Within-household infection events in class $a$ occur at a rate

$$S_a \sum_b \sum_{j=0}^{|\mathcal{I}|} \beta_{\text{int}} \tau_j k_{ab}^{\text{int}} \frac{x_j^b}{(N-1)^d}.$$

Here $\beta_{\text{int}}$ measures the strength of within-household transmission, $\tau_j$ is the infectivity of individuals in the $j$th infectious compartment (i.e. the $j$th compartment to appear in $\mathcal{I}$), and $k_{ab}^{\text{int}}$ is the mean proportion of each unit of time which class $a$ individuals spend exposed to class $b$ individuals within their own household. The infectivities $\tau_j$ are dimensionless quantities reflecting the relative amount of infection generated by individuals in each infectious compartment. In each of the examples in this paper, we choose a scaling so that the (symptomatic) infectious class $I$ has a relative infectivity of 1. The average contact times are stored within a within-household social contact matrix $\boldsymbol{K}^{\text{int}}$ with $(a, b)$th entry $k_{ab}^{\text{int}}$. The exponent $d$ determines the level of density dependence in the within-household contact structure; when $d = 0$ within-household contacts are entirely density dependent (risk of within-household infection increases with household size), and when $d = 1$ the within-household contacts are entirely frequency dependent (risk of within-household infection is independent of household size).

**Between-household transmission.** Between-household transmission events in risk class $a$ occur at a rate

$$S_a \sum_b \sum_{j=0}^{|\mathcal{I}|} \beta_{\text{ext}} \tau_j k_{ab}^{\text{ext}} \langle x_j^b \rangle, \qquad (5)$$

where $k_{ab}^{\text{ext}}$ is a between-household social contact rate taken from the between-household contact matrix $\boldsymbol{K}_{\text{ext}}$ and $\langle x_j^b \rangle$ is the proportion of individuals belonging to risk class $b$ who are currently in epidemiological compartment $j$. Using Eq 3, this is a time-dependent quantity given by

$$\frac{\langle x_j^b(t) \rangle}{\langle N_b \rangle} = \frac{\sum_k x_j^b(k) H_k(t)}{\langle N_b \rangle}.$$

The estimation of the between-household transmission strength $\beta_{\text{ext}}$ is covered in Section 1.

The external infection rates are the only time-dependent component of the household dynamics, and depend on the global state of the population of households.

Throughout this study we use the contact matrix estimates derived by Prem *et al.* [43]. We parameterise $\boldsymbol{K}^{\text{int}}$ as their "Home" coded estimate for the UK, while for $\boldsymbol{K}^{\text{int}}$ we subtract the "Home" coded estimate from the "All locations" estimate to give us a matrix corresponding to non-household contacts. These estimates were calculated prior to the COVID-19 pandemic and thus may not reflect the changes induced by NPI's. Although a follow-up study taking these changes into account has since been published [47], in this study we use the original pre-pandemic estimates because our models aim to make mechanistic forward projections of the impact of NPI's on the baseline pre-pandemic contact behaviour.

**The SEPIR model.** The basic compartmental structure used in this study is the SEPIR model, but our software implementation of the model allows for arbitrary compartmental structures. In this model, when a susceptible individual is infected they enter the exposed compartment, during which the infection incubates but the host is not yet infectious. From here they enter a prodromal phase, during which they are infectious but do not show symptoms. Once symptoms develop they enter the infectious class, and enter the recovered class once they are no longer infectious. Prodromal and symptomatic/fully infectious individuals are assumed to transmit infection with differing intensities. The state of a household stratified into $K$ risk classes is given by

$$(S_1, E_1, P_1, I_1, R_1, \ldots, S_K, E_K, P_K, I_K, R_K).$$

Because we assume individuals can not move between risk classes (i.e. there is no aging in our model) and that each event involves only a single individual, each state transition can be expressed in terms of its impact on a single risk class. The transition rates of the within-household events for this model are as follows:

$$(S_a, E_a, P_a, I_a, R_a) \quad \rightarrow (S_a - 1, E_a + 1, P_a, I_a, R_a) \text{ at rate}$$
$$S_a \beta_{\text{int}} \sum_b k_{ab}^{\text{int}} \frac{1}{N_b^\delta} (\tau_P \langle P_b \rangle + \langle I_b \rangle) \tag{6}$$

$$(S_a, E_a, P_a, I_a, R_a) \rightarrow (S_a, E_a - 1, P_a + 1, I_a, R_a) \text{ at rate } \alpha_E E_a \tag{7}$$

$$(S_a, E_a, P_a, I_a, R_a) \rightarrow (S_a, E_a, P_a - 1, I_a + 1, R_a) \text{ at rate } \alpha_P P_a \tag{8}$$

$$(S_a, E_a, P_a, I_a, R_a) \rightarrow (S_a, E_a, P_a, I_a - 1, R_a + 1) \text{ at rate } \gamma I_a. \tag{9}$$

Here $\tau_P$ is relative infectiousness of prodromal cases compared to symptomatic cases, $\alpha_E$ is the rate at which an infected individual becomes infectious so that $1/\alpha_E$ is the expected latent period, $\alpha_P$ is the rate at which a prodromal individual develops symptoms, so that $1/\alpha_P$ is the expected prodrome period and $(1/\alpha_E + 1/\alpha_P)$ is the expected latent period, and $\gamma$ is the recovery rate of symptomatic infections, so that $1/\gamma$ is the expected symptomatic period, $(1/\alpha_P + 1/\gamma)$ is the expected infectious period, and $(1/\alpha_E + 1/\alpha_P + 1/\gamma)$ is the expected period from infection to recovery.

External infection events in risk class $a$ take place at a rate $\Lambda_a(\mathbf{H}(t))$, which we obtain by substituting our SEPIR compartmental structure into the formula for external infection rates given in Eq 5:

$$\Lambda_a(\mathbf{H}(t)) = S_a \beta_{\text{ext}} \sum_b k_{ab}^{\text{ext}} (\tau_P \langle P_b \rangle + \langle I_b \rangle).$$

Here $\langle P_b \rangle$ and $\langle I_b \rangle$ are the population-level prevalences of prodromal and symptomatic infectious cases respectively in risk class $b$.

**The SEIR model.** Our model of short-term alleviation of NPIs allows for large household bubbles which generate substantially more states in our Markov chain structure than the smaller single households or support bubbles used in our other policy models. To reduce the computational intensity of this model we replace the SEPIR structure with a SEIR structure. This structure combines the prodromal and symptomatic infectious compartments from the SEPIR structure into a single infectious compartment containing all individuals who are currently shedding virus. As in the SEPIR model, exposed and recovered individuals can not generate new infections, so the only compartment to contribute to the infection rates is the infectious compartment. The transition rates of the within-household events for age class $a$ are given by

$$(S_a, E_a, I_a, R_a) \rightarrow (S_a - 1, E_a + 1, I_a, R_a) \text{ at rate } S_a \beta_{\text{int}} \sum_b k_{ab}^{\text{int}} \frac{\langle I_b \rangle}{N_b^\delta} \tag{10}$$

$$(S_a, E_a, I_a, R_a) \rightarrow (S_a, E_a - 1, I_a + 1, R_a) \text{ at rate } \alpha E_a \tag{11}$$

$$(S_a, E_a, I_a, R_a) \rightarrow (S_a, E_a, I_a - 1, R_a + 1) \text{ at rate } \gamma I_a, \tag{12}$$

where $\alpha$ is the rate at which exposed individuals develop infection and $\gamma$ is the rate at which

infectious individuals recover. External infection events in risk class $a$ occur at rate

$$\Lambda_a(\mathbf{H}(t)) = S_a \beta_{\text{ext}} \sum_b k_{ab}^{\text{ext}} \langle I_b \rangle,$$

where $\langle I_b \rangle$ is the population-level prevalence of infectious cases in risk class $b$.

## Estimation of within-household mixing parameters

We parameterise the within-household transmission strength $\beta_{\text{int}}$ and the density parameter $d$ using estimates of susceptible-infectious transmission probability (SITP). This SITP is the probability that an average individual in an average household acquires infection from a single index case in their own household at some point during that index case's lifespan. For $d > 0$ the rate of transmission across any given pair will decrease with household size, and so estimates of SITP are typically stratified by household size. The continuous-time Markov chain structure of our model means that infections occur as the points of a Poisson process, so that the SITP is equivalent to the probability that at least one event occurs in this Poisson process. For compartmental structures with multiple infectious compartments, this Poisson process will be divided into multiple stages with different associated rates and expected durations. If the index case in a household of size $N$ spends $t_j$ units of time in the $j$th infectious compartment (i.e. the $j$th compartment to appear in $\mathcal{I}$), then the probability that zero events happen in this Poisson process is given by

$$\exp\left(-\beta_{\text{int}} \frac{1}{N^\delta} T_{\text{int}} \sum_{j=0}^{|\mathcal{I}|} \tau_j t_j\right). \tag{13}$$

Here $T_{\text{int}}$ is the expected time which two individuals chosen uniformly at random from the same household (itself chosen uniformly at random from the household composition distribution) spend in contact with each other per unit time. This is the leading eigenvalue of the matrix

$$NK^{\text{int}},$$

where $N$ is a diagonal matrix with $i$th diagonal entry $\langle N_i \rangle / \langle N \rangle$, so that the product $NK^{\text{int}}$ gives the within-household contact matrix $K^{\text{int}}$ with each row scaled by the proportion of the population belonging to the corresponding risk class.

The SITP for a household of size $N$, which we will denote by $p_N$, is given by integrating Eq 13 over the exponentially distributed durations $t_j$ and subtracting from 1 to give the probability that at least one event occurs:

$$p_N = 1 - \int_0^\infty \dots \int_0^\infty \exp\left(-\beta_{\text{int}} \frac{1}{N^\delta} T_{\text{int}} \sum_{j=0}^{|\mathcal{I}|} \tau_j t_j\right) \prod_{j=0}^{|\mathcal{I}|} \gamma_j \exp\left(-\gamma_j t_j\right) \mathrm{d}t_0 \dots \mathrm{d}t_{|\mathcal{I}|} \tag{14}$$

$$= 1 - \prod_{j=0}^{|\mathcal{I}|} \gamma_j \int_0^\infty \dots \int_0^\infty \exp\left(-\sum_{j=0}^{|\mathcal{I}|}\left(\beta_{\text{int}} \frac{1}{N^\delta} T_{\text{int}} \tau_j + \gamma_j\right) t_j\right) \mathrm{d}t_0 \dots \mathrm{d}t_{|\mathcal{I}|} \tag{15}$$

$$= 1 - \prod_{j=0}^{|\mathcal{I}|} \gamma_j \frac{1}{\left(\frac{1}{N^\delta} \beta_{\text{int}} T_{\text{int}} \tau_j + \gamma_j\right)} \tag{16}$$

$$= 1 - \prod_{j=0}^{|\mathcal{I}|} \frac{1}{\frac{1}{N^\delta} (\beta_{\text{int}}/\gamma_j) T_{\text{int}} \tau_j + 1}. \tag{17}$$

**Table 2. Estimated susceptible-infectious transmission probability by household size from House *et al.* [30].**

| Household size, $N$ | 2 | 3 | 4 | 5 | 6 |
| --- | --- | --- | --- | --- | --- |
| SITP, $p_N$ | 0.345 | 0.274 | 0.230 | 0.200 | 0.177 |

Here $\gamma_j$ is the rate at which individuals leave the $j$th infectious compartment, so $1/\gamma_j$ gives the expected time spent in this compartment. Given a set of size-stratified estimates of SITP, we estimate $\beta_{\text{int}}$ and $d$ by fitting the formula in Eq 14 using least squares. In our software implementation we perform this fitting using the `minimize` function from Scipy's `optimize` library [48]. All of our SITP estimates are taken from a previous analysis of data from the Office for National Statistics (ONS) COVID-19 Infection Survey [30]. This analysis divides the survey data into four tranches corresponding to different periods in the evolution of the UK's COVID-19 epidemic. For our fitting we use the estimates of SITP from Tranche 2 of the data, running from September 1st to November 15th 2020, during which there was high prevalence of infection and minimal presence of variant strains. These estimates are quoted in Table 2. The estimated SITP decreases with household size, although at a slower rate than would be expected under purely density-dependent transmission (where doubling household size would halve the SITP), suggesting an intermediate level of density dependence.

## Estimation of growth rate from model parameters

In this section we introduce an Euler-Lotka equation approach for estimating the early exponential growth rate of cases from the model parameters. This allows us to estimate the impact of NPIs on the growth in cases when few people in the population are immune. The Euler-Lotka equation we derive is linear in the between-household transmission rate, which gives us a convenient method for estimating this transmission rate given an estimate of the early exponential growth rate. For ease of notation, in this section we will assume we are working in either an SEIR or SEPIR compartmental framework.

We start by introducing the concept of *household outbreak type*. A household containing at least one exposed, prodromal, or infectious individual is said to be experiencing a household outbreak of type $i$ if the household composition is given by $\mathbf{N}(i)$ and the index case which triggered the current outbreak was of class $C_{k(i)}$. The functions $\mathbf{N}(i)$ and $C_{k(i)}$ are defined such that each $i$ corresponds to a unique combination of household composition and index case class, and we only consider combinations such that $N_{k(i)}(i) > 0$, since if this is not the case there will be no members of risk class $C_{k(i)}$ present to trigger the outbreak. Denote the number of household outbreak types by $M$. Let $\boldsymbol{\lambda}(t)$ be an $M$ by $M$ time-dependent matrix whose $(i, j)$-th entry $\lambda_{ij}(t)$ gives the expected rate at which a household experiencing an outbreak of type $i$ which started at time 0 generates new outbreaks of type $j$ at time $t$. Finally, we define $\mathbf{I}(t)$ to be a time-dependent $M$-dimensional vector whose $i$th entry $I_i(t)$ is the number of households currently experiencing a household outbreak of type $i$. If the early exponential growth rate of the population-level epidemic is $r$, then $\mathbf{I}$ will obey the following Euler-Lotka equation:

$$\mathbf{I} = \mathbf{I} \int\limits_0^\infty \boldsymbol{\lambda}(t) e^{-rt} \mathrm{d}t.$$

The rate $\lambda_{ij}(t)$ is given by

$$\lambda_{ij}(t) = \mathbf{H}^i(t) \boldsymbol{F} \boldsymbol{C}$$

where $\mathbf{H}^i(t)$ is the state distribution of a household of type $i$ at time $t$, $\boldsymbol{F}$ is an $L$ by $K$ matrix

whose $(s, c)$th entry is the rate at which a household in state $s$ generates cases in class $c$, and $\mathbf{C}$ is a $K$ by $M$ matrix whose $(c, i)$th entry gives the probability that an infectious case of age $c$ triggers a within-household outbreak of type $i$. The rate $F_{sc}$ is given by

$$F_{sc} = \langle N_c \rangle \sum_d \sum_{j \in \mathcal{I}} \beta_{\text{ext}} \beta_j K_{cd}^{\text{ext}} x_d^j(s),$$

where $\langle N_c \rangle$ is the proportion of the population who are in risk class $c$, $x_d^j(s)$ is the number of individuals of class $d$ and epidemiological compartment $j$ in a household in state $s$, $\beta_j$ is the relative infectivity of compartment $j$, and $\beta_{\text{ext}}$ is the between-household transmission rate. From this formula, we can decompose $\mathbf{F}$ as $\mathbf{F} = \beta_{\text{ext}} \tilde{\mathbf{F}}$, a fact which we will use later in the estimation of $\beta_{\text{ext}}$. From the definition of household outbreak type, $C_{ci}$ is equal to the probability that a class $c$ individual belongs to a household of composition $\mathbf{N}(i)$ if type $i$ households have index case class $C_{k(i)} = c$, and zero otherwise. Assuming that repeated imports into the same household are vanishingly rare in the early stages of the population-level outbreak, the evolution of a within-household outbreak will depend only on the internal dynamics of the household, and so

$$\mathbf{H}^i(t) = \mathbf{H}_0^i \exp(t\mathbf{Q}_{\text{int}}), \tag{18}$$

where $\mathbf{H}_0^i$ is a point distribution centered on the initial state of type $i$ households. For the SEIR and SEPIR compartmental structures, this index state has $S_c = N_c$ for $c \neq k(i)$, and $S_c = N_c - 1$, $E_c = 1$ for $c = k(i)$, with all other entries of the state vector set to zero. If we denote the position of this index state in our list of states by $s_i$, we can define a matrix $\mathbf{H}_0$ with $(i, s_i)$th entry equal to 1 and all other entries equal to zero for $i = 1, \ldots, M$, which maps each outbreak type to its corresponding initial state. Then our Euler-Lotka equation is given by

$$\mathbf{I} = \mathbf{I}\mathbf{H}_0 \int_0^\infty \exp(t(\mathbf{Q}_{\text{int}} - r\mathbf{I})) \mathrm{d}t \mathbf{F}\mathbf{C}.$$

The household outbreak type profile during the early evolution of the population-level epidemic is thus given by the eigenvalue-1 eigenvector of

$$\mathbf{H}_0 \int_0^\infty \exp(t(\mathbf{Q}_{\text{int}} - r\mathbf{I})) \mathbf{F}\mathbf{C} \mathrm{d}t.$$

To estimate the growth rate $r$ from model parameters, we apply a root-finding algorithm to the function

$$f(x) = \lambda - 1,$$

where $\lambda$ is the leading eigenvalue of the matrix

$$\mathbf{H}_0 \int_0^\infty \exp(t(\mathbf{Q}_{\text{int}} - x\mathbf{I})) \mathbf{F}\mathbf{C} \mathrm{d}t.$$

On the other hand, given an empirical growth rate $r$, we can estimate $\beta_{\text{ext}}$ using the decomposition $\mathbf{F} = \beta_{\text{ext}} \tilde{\mathbf{F}}$. If we define $\tilde{\lambda}$ to be the leading eigenvalue of

$$\mathbf{H}_0 \int_0^\infty \exp(t(\mathbf{Q}_{\text{int}} - r\mathbf{I})) \tilde{\mathbf{F}}\mathbf{C} \mathrm{d}t, \tag{19}$$

then

$$\beta_{\text{ext}} = 1/\tilde{\lambda}. \tag{20}$$

To calculate the integral $\int_0^\infty \exp(t(\mathbf{Q}_{\text{int}} - r\mathbf{I}))\mathbf{FC}\mathrm{d}t$, we use the results of Pollett and Stefanov on path integrals for Markov chains [49]. We can interpret this time-discounted integral as describing a process identical to our within-household dynamics, but with a constant flow of probability at rate $r$ to an absorbing null state which has no other effect on the dynamics. If we interpret $\mathbf{V} = \mathbf{FC}$ as an $M$-dimensional household outbreak type-stratified reward function, so that the $s$th row tells us the per-unit-time reward vector accumulated while in state $s$ and the $i$th column of the integral is precisely the expected number of household outbreaks of type $i$ generated during the lifetime of a given household outbreak, stratified by outbreak type. Using the results of Pollett and Stefanov [49], this expected reward by type is given by the solution $\mathbf{y}$ to the system of linear equations

$$(r\mathbf{I} - \mathbf{Q}_{\text{int}})\mathbf{y} = \mathbf{V}_i,$$

where $\mathbf{V}_i$ is the $i$th column of $\mathbf{V}$. This reduces the integral calculation to a series of linear solves.

## Construction, initialisation, and solution of the self-consistent equations

At the beginning of our model description we introduced the self-consistent equations (Eqs 1 and 2) which determine the evolution of an epidemic in a population of households. We now outline the construction and solution of this system of equations from the rate equations we have defined above and real-world household composition data.

Suppose that we have a list of all the household compositions which appear in the population, along with the proportion of all households which have that composition. Data in this format is typically available from census agencies, although it may require some processing; the UK's ONS, for instance, offers this information at a fine-grained geographical scale [50, 51], but this needs to be aggregated in order to calculate a country-level distribution. Let $h_{\mathbf{N}}$ be the proportion of households in composition $\mathbf{N} = (N_1, \ldots, N_K)$ with the set of households of size $N$ being

$$\mathcal{H}(N) = \left\{ \mathbf{N} \middle| \sum_i N_i = N \right\}.$$

Then the proportion of households of size $N$ is

$$h_N = \sum_{\mathbf{N} \in \mathcal{H}(N)} h_{\mathbf{N}}.$$

Because very large households are rare in most populations while carrying a high computational cost in our model, we build our model population using a truncated version of the household composition distribution where the largest households are removed. Letting $N_{\text{max}}$ be the maximum observed household size, the proportion of individuals belonging to households of size $N$ or larger is given by

$$h_{\geq N} = \frac{\sum_{n=N}^{N_{\text{max}}} N h_N}{\sum_{n=1}^{N_{\text{max}}} N h_N}.$$

Then, for instance, to remove the top 5% of the population by household size from the household composition distribution, we just find the smallest value of $N$ for which $h_{\geq N} < 0.05$, remove all the compositions with $\sum_{i=1}^{K} N_i \geq N$, and then renormalise what remains of the distribution. Performing this truncation on the 2011 UK census data, we find that 95% of the population belongs to households of size six or smaller.

The transition matrix for our household-stratified epidemic model is constructed block by block. For each composition $\mathbf{N}$ observed in the data, we construct a transition matrix $\mathbf{Q_N}$ which describes the household-level epidemic process of a household in that composition. For the household composition distribution of our model population to match that of the real population, we need the total probability corresponding to each block of $\mathbf{Q}$ to be equal to the proportion of households in the corresponding composition. Because there are no off-block-diagonal elements of $\mathbf{Q}$ (the composition of a household does not change), the probability corresponding to each block will not change as we solve the master equations, and so it is sufficient to choose an initial probability vector $\mathbf{H}_0$ which meets this requirement. This can be achieved by cycling over each household composition $\mathbf{N}$, defining a conditional distribution for households in composition $\mathbf{N}$ and then scaling by $h_{\mathbf{N}}$. However, there are a few standard starting conditions which we will explain here.

To initialise the model with an entirely susceptible population, we assign probability $h_{\mathbf{N}}$ to the state

$$N_1, 0, 0, 0, 0, N_2, 0, 0, 0, 0, \ldots, N_K, 0, 0, 0, 0)$$

with $S_a = N_a$ for each risk class $a$. Thus if we denote the $k$th state in our list of states by $(S_1^k, \ldots, R_K^k)$, with composition $\mathbf{N}(k) = (N_1(k), \ldots, N_K(k))$, then we define $\mathbf{H}_0$ so that

$$H_0^k = \begin{cases} p_{\mathbf{N}(k)} & \text{if } S_i^k = N_i(k) \text{ for } i = 1, \ldots, K \\ 0 & \text{otherwise.} \end{cases}$$

A simulation initiated with this zero-infection state will have completely static dynamics, but if the infection events in our model are modified to include imports of infection from outside of the population then this initial condition can be used to simulate the beginning of an epidemic.

To initialise the model with infection present, we need to specify not only the relative sizes of each epidemiological compartment, but also the distribution of epidemiological status by age class and household composition. Choosing an arbitrary distribution can create a pathological starting condition in regions of phase space which are far away from the model's usual behaviour. This results in a long "burn in" period when solving the self-consistent equations, during which the model state moves towards a more natural region of phase space. Because cases and acquired immunity will accumulate during this burn-in period, a long burn-in makes it difficult to simulate a realistic epidemic starting from specific infectious prevalence and background immunity levels. While specific applications may call for their own distributions, in all of the examples presented in this paper we use an eigenvector approach which attempts to find a distribution of cases close to the "natural" distribution during the early growth of an epidemic. This approach aims to minimise the burn-in period so that epidemics with specific starting infectious prevalence and background immunity levels can be simulated.

We will make the simplifying assumption that in our initial conditions all households are in one of three points in their infectious history: (1) completely naive to infection with all individuals susceptible; (2) at the very beginning of a within-household outbreak with exactly one individual exposed and all other individuals susceptible; or (3) at the end of a within-household outbreak with all individuals either susceptible or recovered. We will define our initial

conditions by defining distributions of households at points (2) and (3) in their infectious history, and then defining corresponding proportions of all-susceptible households so that we obtain the correct household composition distribution. In the derivation of our growth rate estimation formula in § above, we argued that if the early exponential growth rate of cases is $r$, then we need to consider the leading eigenvalue of the matrix

$$\boldsymbol{H}_0 \int_0^\infty \exp(t(\boldsymbol{Q}_{\text{int}} - r\boldsymbol{I}))\boldsymbol{FC}\mathrm{d}t,$$

and associated unit eigenvector $\mathbf{I}$. This leading unit eigenvector defines the distribution of within-household outbreaks by household outbreak type, so that the $i$-th element $I_i$ tells us what proportion of all households experiencing an outbreak during the early exponential growth phase are in composition $\mathbf{N}(i)$ with an index case in risk class $C_{k(i)}$. For each household outbreak type $i$, define $\mathbf{x}_0^i$ to be the starting state of such a household outbreak, i.e. the state with all household members susceptible apart from one individual of risk class $C_{k(i)}$ which belongs to whichever is the "first" infectious class in the compartmental structure; in the SEIR and SEPIR models this is the exposed class. We can define an initial condition close to the early exponential growth regime of the system by assigning a small amount of probability to each of these starting states, with the amount of probability assigned to the $i$th such state directly proportional to $I_i$. Let $I_0$ be the desired initial prevalence. If we set this constant of proportionality to be $I_0\langle N\rangle$ so that the probability of being in the $i$th starting state is

$$H_{\mathbf{x}_0^i} = I_i I_0 \langle N\rangle$$

then the prevalence across these starting states will be

$$\frac{\sum_i I_i I_0 \langle N\rangle}{\langle N\rangle} = I_0,$$

since $\mathbf{I}$ was defined to be a unit eigenvector. Denote by $\mathbf{H}_0^I$ this initial profile of newly infected households. To add households which have already experienced an outbreak to our initial state, we need to calculate the state distribution of these households. Assuming no repeat infections, their dynamics are given by Eq 18 and so we can get the distribution of post-outbreak states by solving this system with initial condition $\mathbf{H}_0^I / \parallel \mathbf{H}_0^I \parallel$, the distribution of starting states of a within-household outbreak, conditioned on being at the start of such an outbreak. Thus, denoting by $H_0^R$ the initial profile of post-outbreak households, we have that

$$\mathbf{H}_0^R \propto \int_0^\infty \exp(t\mathbf{Q}_{\text{int}}) \frac{\mathbf{H}_0^I}{\parallel \mathbf{H}_0^I \parallel} \mathrm{d}t.$$

   This indefinite integral is not particularly easy to calculate, but since without reinfections the state distribution should converge in finite time to a post-outbreak state distribution, we can approximate the integral by solving Eq (18) forward for some period substantially longer than the expected duration of a within-household outbreak using an ODE solver. In our code, we set this period to be one year. This solution is a unit vector $\tilde{\mathbf{H}}_0^R$. To initialise our model with a desired initial immunity level $R(0)$, we estimate the proportion of individuals in a population

with state distribution $\tilde{\mathbf{H}}_0^R$, which we denote $\tilde{R}(0)$, and then set

$$\mathbf{H}_0^R = \frac{R(0)}{\tilde{R}(0)}\tilde{\mathbf{H}}_0^R.$$

This gives us an initial profile of post-outbreak households with a chosen starting immunity level and the distribution across age classes and household compositions which we expect to arise from our infectious disease dynamics. The remaining households in our initial state distribution will be entirely susceptible. For each household composition there is precisely one such state, and we require that the initial state distribution has the population's empirical composition distribution. Define $\mathbf{S}(\mathbf{N}_i)$ to be the unique state with all members of a household of composition $\mathbf{N}_i$ susceptible. We can define an initial profile $\mathbf{H}_0^S$ of all-susceptible households as follows:

$$H_0^S(\mathbf{S}(\mathbf{N}_i)) = p(\mathbf{N}_i) - \sum_{\{\mathbf{x}:\mathbf{N}(\mathbf{x})=\mathbf{N}_i\}} (H_0^I(\mathbf{x}) + H_0^R(\mathbf{x})).$$

The initial state distribution

$$\mathbf{H}_0 = \mathbf{H}_0^S + \mathbf{H}_0^I + \mathbf{H}_0^R$$

has the correct state distribution as well as the chosen initial prevalence and background immunity.

Given a starting state distribution $\mathbf{H}_0$, the state distribution over the time period $(t_0, t_{\text{end}})$ is calculated by solving the nonlinear Equations Eq 14 with initial conditions $\mathbf{H}_0$ over this time period. Using formulae along the lines of Eq 1, we can derive projections of quantities like expected infectious prevalence or population-level immunity over time, which we can also stratify according to household composition or risk class.

In Fig 1 we plot the early growth in infections and an exponential curve on the same logarithmically scaled axes to demonstrate that the model with our chosen initial conditions converges to exponential growth within a few days of "in-model" time. The exponential curve is given by $f(t) = (\langle E(14)\rangle + \langle P(14)\rangle + \langle I(14)\rangle)e^{r(t-14)}$, so that it is calibrated with the exponential growth in cases on day 14 of the simulation. Comparing the two curves demonstrates that simulations starting from our chosen initial conditions settle into an exponential growth phase within a few days of model time. In Section 2 of the Supporting Information S1 Appendix we perform a similar comparison for model simulations initialised with varying levels of background immunity and show that the early dynamics are close to exponential growth for populations with a background immunity of up to 10%.

## Household composition distributions

Data on age-stratified household compositions in England and Wales gathered during the 2011 census is publicly available from the ONS as datasets CT1088 (households of size six and under, split across thirteen files organised by geographical location) and CT1089 (households of size seven or more) [50, 51]. This data lists every household surveyed in the 2011 UK census with the number of individuals present belonging to each age 10 year age band from age 0 to 100. We merged the first two age classes into a single 0–19 age class and the rest into a 20+ age class to give us a list of households in terms of this simplified age structure. We extracted a list of all the household compositions observed according to this two-class age structure, and the proportion of households in each composition. The dataset includes household compositions which are large and comparatively rare. These large households introduce a large number of extra dimensions into our mechanistic model, while their comparative rarity means limited

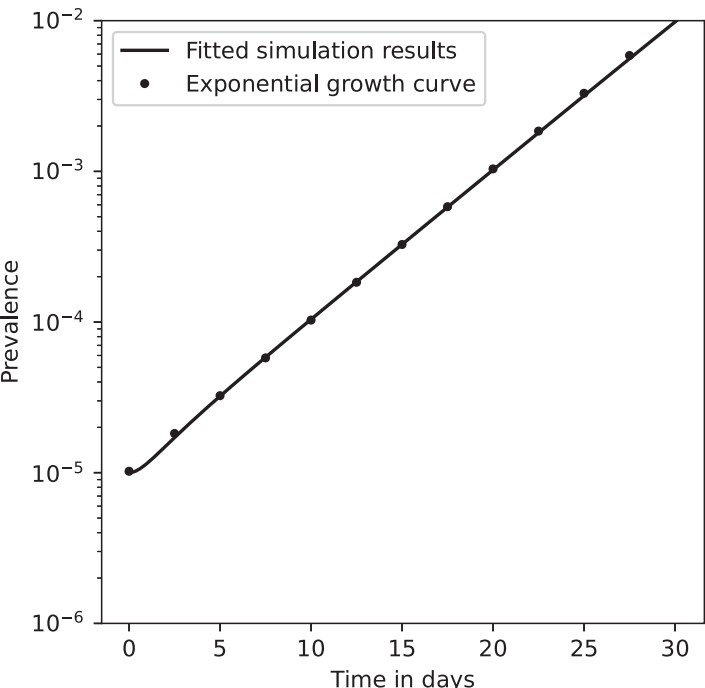

**Fig 1. Convergence of model from initial conditions to exponential growth regime.** Our model is simulated from initial conditions with starting prevalence $10^{-5}$ and no background immunity. The exponential curve is callibrated to the prevalence 15 days from the start of the simulation.

realism is lost by removing them from our model population. With this in mind, we use a truncated version of the empirical household composition distribution with all compositions of size seven and over removed. Households of size six or less account for 98.2% of the households in England and Wales and contain 97.8% of their combined population and so this truncation should result in a minimal loss of accuracy.

For our out-of-household isolation (OOHI) analysis, we incorporate a third class of clinically vulnerable individuals into the population so that each household composition specifies the number of children, less clinically vulnerable adults, and clinically vulnerable adults present in the household. To estimate the household composition distribution under this tripartite division, we combine the CT1088 dataset with estimates, publicly available from the ONS and listed here in Table 3, of the number of individuals shielding from the 9th to the 16th of July

**Table 3. Estimated age cohort populations and numbers shielding from ONS data [52, 54], estimated age distribution of shielding population, and proportion of each age group shielding.**

| Age class | Est. population | Est. shielding | % of all shielding | % of age group shielding |
|---|---|---|---|---|
| 00–20 | 13282321 | 100000 | 4.04 | 0.75 |
| 20–29 | 7289272 | 79000 | 4.04 | 1.08 |
| 30–39 | 7541596 | 123000 | 5.05 | 1.63 |
| 40–49 | 7130109 | 183000 | 8.08 | 2.57 |
| 50–59 | 7578112 | 339000 | 15.15 | 4.47 |
| 60–69 | 5908575 | 445000 | 20.20 | 7.53 |
| 70–74 | 2779326 | 298000 | 13.13 | 10.72 |
| 75+ | 4777650 | 668000 | 30.30 | 13.98 |

2020 [52]. Under UK guidance, "shielding" is defined to be a voluntary protection measure targeted towards clinically extremely vulnerable individuals [53]. Under the guidance provided in June and July 2020 shielding individuals were advised not to leave their home apart from for daily exercise in open space, with no visitors allowed other than nurses or support or care workers. This is intended to minimise the exposure of clinically extremely vulnerable individuals to infection, with a particular focus on infection outside of their own household. Because of the relatively low proportion of under-20's in the shielding data, we assume in our model that only adults shield, and so our clinically vulnerable class specifically corresponds to clinically vulnerable adults. Our estimation assumes that each individual of age class $C_i$ in the ten-year age band structure has an independent probability of belonging to the clinically vulnerable class equal to the proportion of individuals in the $C_i$ class who are shielding. From the census household sample, we calculate the proportion of households in each ten-year-age-band-stratified composition. For each composition in this distribution, we use the shielding probabilities to generate the set of compositions and corresponding probabilities which arise when one or more members of the household belong to the clinically vulnerable class, while adding a zero to the end of the original composition corresponding to the empty vulnerable class, and multiplying its probability by the probability that no members of the household are clinically vulnerable. Doing this for each composition in the original distribution gives us a list of household compositions with ten year age bands plus a clinically vulnerable class, along with a corresponding composition distribution. This list will have repeats, since certain compositions which are distinct under the census stratification will be identical when certain household members are moved to the clinically vulnerable class (for instance, a household containing a single clinically vulnerable 25 year old will be identical to one containing a single clinically vulnerable 75 year old under the ten-year-age-bands-plus-clinically-vulnerable stratification). We therefore add together the probabilities corresponding to each copy of any repeating compositions to give us a distribution of unique compositions. Finally, we merge the first two age classes into a single 0–19 age class and the adult age classes into a single less clinically vulnerable 20+ age class, to give us a list of compositions and proportions under our children-adults-vulnerable stratification. Again, this list of compositions will have repeats, which we remove by adding together the probabilities corresponding to each copy of the repeating compositions.

Our code, implemented with the MATLAB programming language, for calculating household composition distributions is publicly available via GitHub website under an open source Apache 2.0 license [55].

## Results

### Impact of transmission controls at the within- and between-household level

Non-pharmaceutical interventions have mainly focused on reducing between-household transmission through measures such as school and workplace closures and self isolation of detected cases. In this section we model the combined impact of these more conventional between-household controls and measures to reduce within-household transmission. Within-household measures were of particular interest to policy makers in the UK during the emergence of the alpha variant of COVID-19 in early 2021, with an emphasis on communications campaigns designed to inform individuals about steps they could take to minimise transmission within their own household [42]. Practical measures suggested as part of such a campaign included hand hygiene, cleaning surfaces, opening windows to improve ventilation, spending time in separate rooms, and wearing masks within one's own home. Mask-wearing and hand hygiene as measures against within-household transmission of influenza have been the subject

of randomised control trials prior to the COVID-19 pandemic, although these studies suggest that these measures have a limited impact on transmission [5, 6, 56]. Here our goal is to model the population-level impact of a given reduction in within-household transmission, with no explicit consideration of the exact measures which could bring about such a reduction. The analysis in this section is therefore intended as an assessment whether control of within-household transmission is a worthwhile goal for public health policy makers, as opposed to a direct simulation of any specific within-household control measure.

In this analysis we calibrate the external mixing intensity $\beta_{ext}$ to a doubling time of 3 days (growth rate $r = \log(2)/3$), corresponding to unconstrained spread of wild type COVID-19, using Eqs 19 and 20. We assume that the impact of an NPI targeting within-household transmission acts to reduce the parameter $\beta_{int}$ by a given percentage, and likewise the impact of an NPI targeting between-household transmission acts to reduce the parameter $\beta_{ext}$ by a given percentage. In our analysis we vary these percentage reductions independently to assess the combined impact of within- and between-household interventions. For each combination of percentage reductions we calculate the early exponential growth rate $r$ and solve the self-consistent equations (Eq 2) to generate a 90 day projection of the epidemic from a starting state with no background immunity and a low initial prevalence of 0.001%.

In Fig 2 we present the results of this analysis. Overall, interventions targeting between-household transmission are much more effective than those targeting within-household transmission, with within-household interventions unable to fully control transmission without very intensive reductions in between-household transmission. This is consistent with theoretical results on household-structured models derived by Ball *et al.* [57]. In their study they define a household-level reproductive ratio $R_*$ (although since their study considers more general

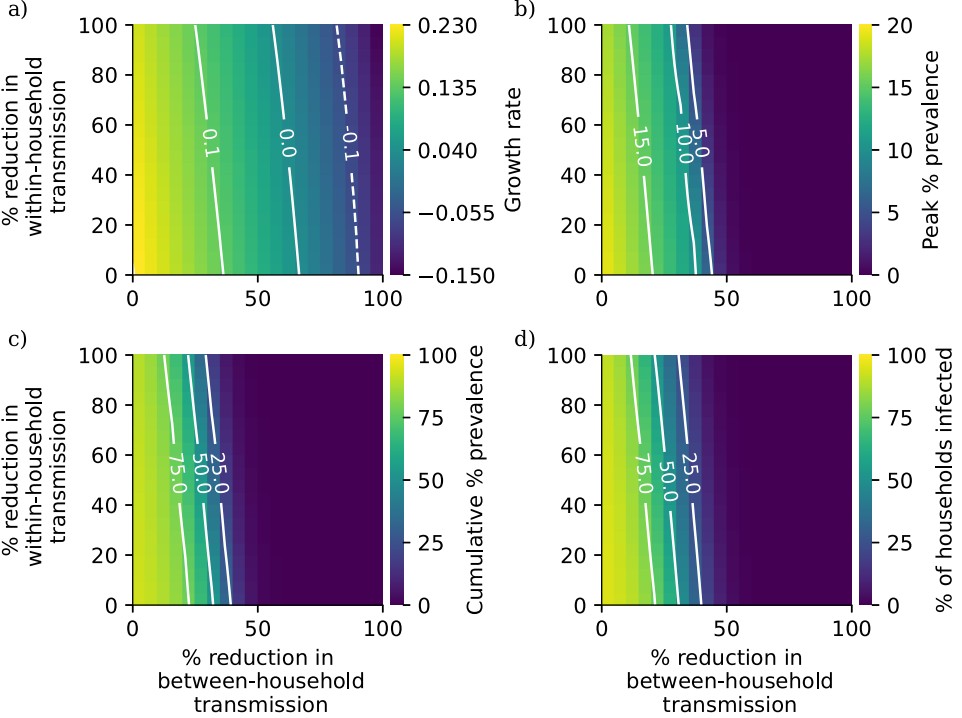

**Fig 2. a)Growth rate, b) peak percentage prevalence, c) cumulative percentage prevalence, d) percentage of households with at least one case during the simulation period as a function of percentage reduction in between-household transmission rates and percentage reduction in within-household transmission rates.**

forms of structure they do not use this terminology), the number of subsequent within-household outbreak generated by each within-household outbreak. This quantity is given by the formula $R_* = R_G \mu$, $R_G$ is the expected number of subsequent infections outside of their own household generated by each case, and $\mu$ is the expected size of a within-household outbreak. The household-level reproductive ratio is a threshold parameter, so that an epidemic is controlled when $R_* < 1$. Because $\mu$ is bounded below by 1 (each within-household outbreak has at least one index case), forcing $R_G$ below one with controls on between-household mixing is always a necessary condition for control of transmission. While the model formulation considered by Ball *et al.* lacks the risk structure we have introduced here, this argument helps to demonstrate why within-household measures are not themselves sufficient to control transmission in our analysis.

Comparing Fig 2b) and 2c) reveals that reducing within-household mixing has a bigger impact on peak prevalence than cumulative prevalence; this suggests that although these measures may not reduce the total number of cases in a given outbreak, they could play a role in reducing pressure on health services whose capacities are defined in terms of hospitalisations at one time rather than over long periods.

As part of this analysis we also calculate the secondary attack ratio by household size. This quantity can be interpreted as the probability that an individual becomes infected, given at least one other member of their household becomes infected. This is distinct from the SITP because it is calculated over the entire duration of a within-household epidemic rather than over the lifetime of the index case. In our model we estimate the secondary attack ratio for households of size $N$ at time $t$ as

$$\sum_{\{k:N(k)=N\}} \frac{R(k) - 1}{N} H_k(t). \tag{21}$$

We make two estimates of this quantity. The first uses our simulation results and takes $t = 90$, so that $H_k(t)$ is the probability that a household is in the $k$th epidemiological state at the end of our simulation. This gives us the expected number of non-index cases per infected household as a fraction of household size in our simulation. These non-index cases may include cases which were infected due to repeated imports of infection and are not traceable back to the first case in their household. For the second estimate, we use $\mathbf{H}_0^R$, the profile of post-outbreak households in the early eigenphase of the epidemic which we calculated during the calculation of the initial conditions. Using this vector in the formula tells us the expected number of non-index cases per infected household as a fraction of household size during a single outbreak with no repeated imports of infection in an average household during the early stages of the epidemic. The first estimate of secondary attack ratio is useful because it can be compared to empirical data from households without knowing explicitly whether all household members were infected in the same local outbreak, whereas the second estimate quantifies the infection's propensity to spread within the household.

In Fig 3 we plot estimates of secondary attack ratio by household size under different levels of reduction in within- and between-household mixing. Fig 3a) and 3b) respectively compare the baseline secondary attack ratio in the absence of interventions with those calculated for 25% and 50% reductions in transmission rates on the within- and between-household levels, as well as on both levels with the same intensity, based on our 90 day simulation output. Fig 3c) and 3d) respectively compare the baseline secondary attack ratio in the absence of interventions with those calculated for 25% and 50% reductions in transmission rates on the within-household level, for a single within-household outbreak with only one import of infection. Without repeated imports of infection, between-household mixing controls will have no

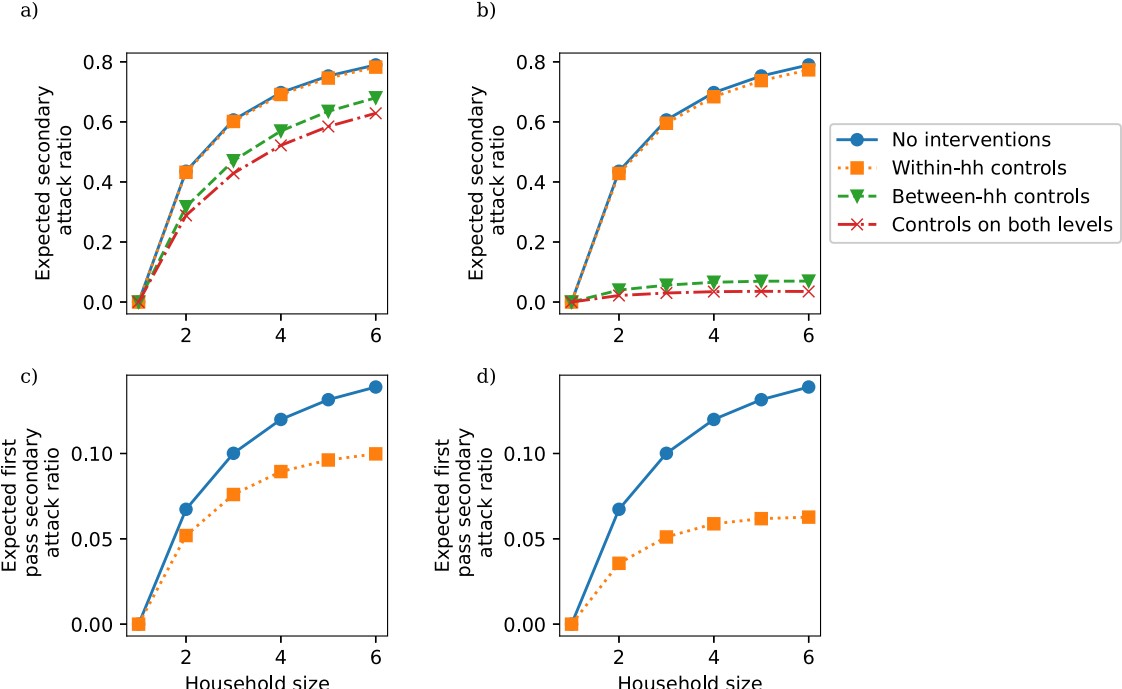

**Fig 3. Expected secondary attack ratios by size.** The first row gives the estimates from the full population dynamics with repeated imports of infection for a) a 25% reduction in transmission from each intervention, and b) a 50% reduction in transmission from each intervention; the second row gives the estimates from a single within-household outbreak without repeated imports of infection for c) a 25% reduction in transmission from each intervention, and d) a 50% reduction in transmission from each intervention.

impact on a within-household outbreak and so there is no need to consider the impact of between-household mixing controls in this case. Fig 3a) suggests that a 25% reduction in within-household mixing will have a minimal impact on the secondary attack ratio, while measures to control between-household mixing have a small but non-negligible impact. While Fig 3c) demonstrates that a 25% reduction in within-household transmission can reduce the secondary attack ratio for a single within-household outbreak, the results in Fig 3a) suggest that this impact will be counteracted by repeated introductions. Fig 3b) shows a similarly limited impact from within-household measures in the absence of between-household measures. Again, comparing Fig 3d) with Fig 3b) suggests that repeated imports of infection substantially reduce the potential impact of within-household transmission reductions. Given the relatively high impact of within-household measures over a single within-household outbreak, our results suggest that within-household measures could be effective in reducing the risk of transmission to particularly vulnerable members of a household in scenarios where imports of infection into the household are already minimised through effective population-measures.

## Lockdown support bubbles

Here we considered the combination of pairs of households into units such as the "support bubbles" presented in guidelines by England's Department of Health and Social Care [58]. Under the policy implemented in England in 2020, a support bubble is defined to be "a support network that links two households", with households belonging to the same bubble functioning as a single household for the purpose of lockdown rules [58]. Only certain households were elligible to form bubbles, including adults living alone, single parents living with children,

and households where only one adult resident did not need continuous care due to disability. Support bubbles represent an exemption to population-level lockdown measures, intended to reduce the challenges these measures pose for members of elligible households. However, these exemptions have the potential to reduce the overall efficacy of lockdown measures by increasing the population-level frequency of social contact and thus increasing the capacity for spread of infection. The possible impact of these measures was a topic of discussion at the UK's Scientific Pandemic Influenza Group on Modelling (SPI-M) in April 2020, where it was argued that members of bubbles would be at greater risk of infection than members of single households, due to the greater routes of potential transmission into a bubble generated by the larger number of bubble members [59]. This argument did not take into account the impact of bubble formation on the total quantity of infection circulating in the population, which will itself impact each bubble or household member's probability of bringing infection into their own bubble or household respectively. Here we use our model to perform a more detailed analysis of the impact of support bubbles, which explicitly accounts for the non-linearities generated by this change in both the structure of a bubble member's local contacts and the total infection circulating in the wider population.

We consider a scenario in the broad spirit of such support bubbling where any household consisting of a single adult plus any number of children (possibly zero) combines with a single other household which can be in any composition. To capture the expected epidemiological impact of long-term bubbling, we compare projections from the SEPIR model applied to the 2011 UK census population to projections from a bubbled version of that population. Explicitly, each household bubble combines the members of a household in composition $(N_1^1, 1)$ or $(N_1^1, 0)$ with the members of a household in composition $(N_1^2, N_2^2)$, where class $C_1$ consists of everyone under 20 and class $C_2$ consists of everyone aged 20 and older. This gives us a bubbled household composition distribution **b**, where if each household which is eligible for bubbling (i.e. contains no more than one adult) bubbles with probability $p_b$, the proportion of post-bubbling households in composition $(N_1, N_2)$ is given by

$$b_{(N_1,N_2)} = \begin{cases} (1-p_b)p_{(N_1,N_2)} & \text{if } N_2 = 0 \\ (1-p_b)p_{(N_1,N_2)} + p_b\sum_{M_1}p_{(M_1,0)}p_{(N_1-M_1,1)} & \text{if } N_2 = 1 \\ p_{(N_1,N_2)} + p_b\sum_{M_1}\left(p_{(M_1,0)}p_{(N_1-M_1,N_2)} + p_{(M_1,1)}p_{(N_1-M_1,N_2-1)}\right) & \text{otherwise.} \end{cases}$$

To analyse the impact of such a bubbling policy, we define a set of self-consistent equations for the SEPIR model in the population of households with support bubbles. We use the Euler-Lotka equations to estimate growth rates in this population under varying levels of adherence to lockdown measures and uptake of support bubble allowances among eligible households, and solve the self-consistent equations to make projections of the corresponding infectious disease dynamics.

In our analysis we vary the uptake of support bubbles among households who are eligible to join one as well as the percentage reduction in between-household transmission due to NPIs. Households are deemed eligible to join a support bubble if they contain at most one adult (again we note that in our model of age structure we take age 20 as the boundary between children and adults). For computational purposes, we do not allow households to form households of size 10 or larger. The transmission rate over within-household contacts, $\beta_{\text{int}}$ is left fixed at the value obtained by fitting to the SITP, listed in Table 1. This reflects the fact that our primary interest here is specifically in modelling the potential detrimental impact of support bubble formation on measures to reduce between-household transmission. For each combination

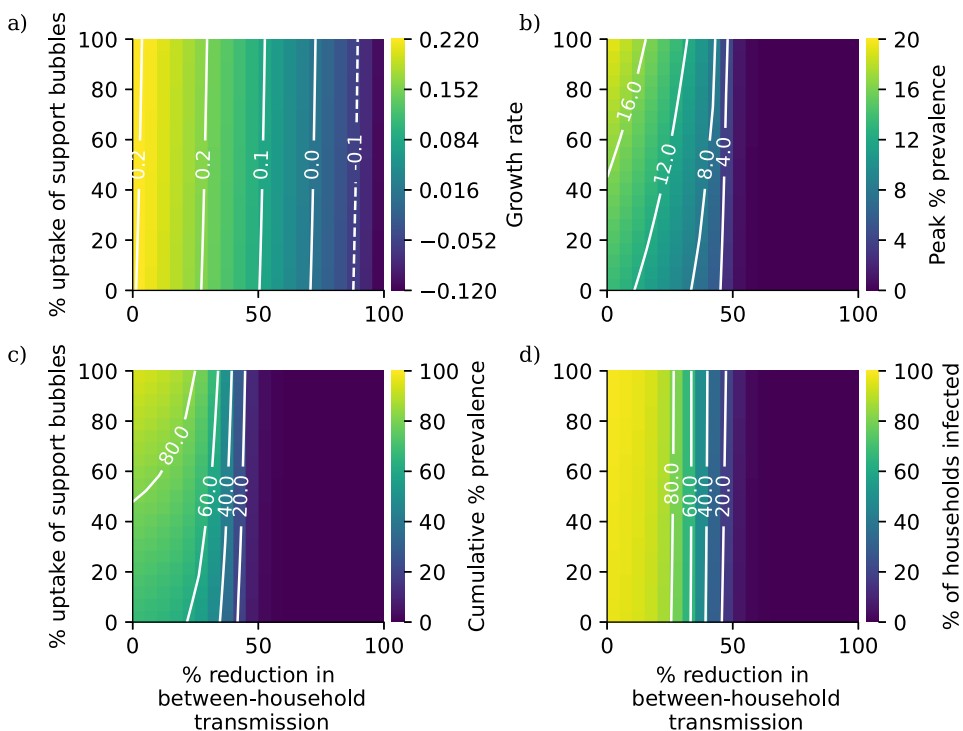

**Fig 4. a)Growth rate, b) peak percentage prevalence, c) cumulative percentage prevalence, d) percentage of households with at least one case during the simulation period as a function of percentage reduction in between-household transmission rates and uptake of support bubbles among eligible households.**

of uptake and transmission reduction we calculate the early exponential growth rate $r$ and solve the self-consistent equations to generate a 90 day projection of the epidemic from a starting state with no background immunity and a low initial prevalence of 0.001%.

The results of the long-term social bubble analysis are presented in Fig 4. Overall we predict that allowing support bubble exemptions will have a limited impact on the infectious disease dynamics. While high levels of uptake are associated with higher peak and cumulative prevalences, the difference in peak prevalence decreases noticeably as we increase the level of reduction in between-household mixing. In particular, allowing support bubbles does not appear to affect the threshold level of between-household controls at which spread of pathogen is controlled. The non-monotonicity observed in Fig 2d) arises from the changes in the household composition distribution that arise from allowing bubbles to form at different uptake rates; initially at low levels of uptake the increasing average household size dominates, with the population concentrated in larger households so that the total proportion of households infected decreases, while at higher levels of uptake the increase in cumulative prevalence dominates.

## Temporary relaxations of NPIs

In contrast to the long-term bubbling considered in the analysis of support bubbles, our model can also be used to study short-term bubble policies, where a few households join together and mix intensively over a short period. Whereas we interpreted long-term bubbles as being associated with exemptions to lockdown measures, short-term bubbles can be interpreted as a feature of relaxations of lockdown measures, where individuals are temporarily allowed to mix outside of their own households before returning to lockdown-period behaviour. An example

of such a policy is given by the *Making a Christmas bubble with friends and family* [60] guidance from the UK Cabinet Office [60]. Under the guidelines which were eventually implemented, mixing was allowed between pairs of households on December 25th 2020, with no other relaxations of or exemptions to the background suite of NPIs which were in place at that time. Prior to the announcement of these guidelines policies were proposed involving mixing between more than two households or over more than one day [61], which were rejected once the extremely high levels of hospitalisation in the UK in December 2020 became apparent. Here we consider the possible impact of a range of relaxation policies involving different numbers of households and different numbers of consecutive or non-consecutive days of mixing by using our model to simulate the dynamics of infection in a population which moves between a baseline population of single households and a bubbled population of small groups of households which mix intensively.

First, note that in the long-term bubbling model, the impact of two households merging can be summarised as follows, using $\oplus$ to represent bubbling in a natural manner:

$$(M_1, M_2) \oplus (N_1, N_2) \rightarrow (M_1 + N_1, M_2 + N_2),$$

so that the bubbled household had epidemiological state

$$(S_1^M + S_2^N, E_1^M + E_2^N, P_1^M + P_2^N, I_1^M + I_2^N, R_1^M + R_2^N),$$

where $X_i^M$ and $X_i^N$ are respectively the number of class $C_i$ individuals in compartment $X$ originally belonging to the first and second households in the bubble.

In contrast, to model short-term bubbling, we need to allow households to enter bubbles and leave them, which means we need to keep track of the number of individuals in a given risk class and compartment in the bubbled household who originally belonged to each of the contributing households. We therefore define our bubbled population in terms of a composition structure which tracks the source household of members of a bubbled household, so that for the two-household example above,

$$(M_1, M_2) \oplus (N_1, N_2) \rightarrow (M_1, M_2, N_1, N_2).$$

A bubble of two households stratified by two classes will have a four-dimensional composition, while a bubble of three households stratified by two classes will have a six-dimensional composition. To prevent our system size from becoming too large in this analysis, we will ignore age-risk structure and model a single age class consisting of all the individuals in the population. This means that in our analysis we are unable to trace the impact of these policies on the burden of disease in specific age classes, although are model does allow for similar analyses with more complex age structures given sufficient computing time.

We can, therefore, represent the short-term bubbling of two households as follows:

$$N_1 \oplus N_2 \rightarrow (N_1, N_2),$$

and similarly for three households:

$$N_1 \oplus N_2 \oplus N_3 \rightarrow (N_1, N_2, N_3).$$

Two-household bubbles will have epidemiological state

$$(\mathbf{x}_1, \mathbf{x}_2) = (S_1, E_1, P_1, I_1, R_1, S_2, E_2, P_2, I_2, R_2),$$

and three-household bubbles will have

$$(\mathbf{x}_1, \mathbf{x}_2, \mathbf{x}_3) = (S_1, E_1, P_1, I_1, R_1, S_2, E_2, P_2, I_2, R_2, S_3, E_3, P_3, I_3, R_3).$$

With this structure, we can then dissolve bubbles into their constituent households at the end of the bubbling period, keeping track of the epidemiological trajectory of their members. In our study we model five different short-term bubble policies which were discussed in the weeks leading up to Christmas 2020, including the policy which was eventually implemented (mixing between a maximum of two households on December 25th only). Each policy consists of a set of days on which bubbling is allowed, plus the number of households which are allowed to form a bubble on those days. We assume that when permitted, every household will join a bubble of the maximum permitted size, so on days when two-household bubbles are allowed the entire population will consist of two-household bubbles with no singletons, and likewise on days when three-household bubbles are allowed the entire population will consist of three-household bubbles with no singletons or two-household bubbles.

The bubbling strategy is implemented by first constructing a bubbled household composition distribution. For a two-household bubble policy, the bubbled compositions $(N_1, N_2)$ and $(N_2, N_1)$ are identical up to the ordering of the constituent households, and so we can construct a "triangular" set of bubbles such that households of sizes $N_1$ and $N_2$ will only form a bubble if $N_1 \geq N_2$. The bubbled composition distribution for the two-household-bubble population is then given by

$$b_{(N_1, N2)} = \begin{cases} p_{N_1}^2 & \text{if } N_1 = N_2 \\ 2p_{N_1} p_{N_2} & \text{otherwise,} \end{cases}$$

with the factor of 2 in the second formula accounting for the fact that we treat $(N_1, N_2)$ and $(N_2, N_1)$ bubbles as identical. If we enforce the condition that $N_1 \geq N_2 \geq N_3$, then the bubbled composition distribution for the three-household-bubble population is given by

$$b_{(N_1, N2, N3)} = \begin{cases} p_{N_1}^3 & \text{if } N_1 = N_2 = N_3 \\ 3p_{N_1}^2 p_{N_3} & \text{if } N_1 = N_2 \neq N_3 \\ 3p_{N_1} p_{N_2}^2 & \text{if } N_1 \neq N_2 = N_3 \\ 6p_{N_1} p_{N_2} p_{N_3} & \text{otherwise,} \end{cases}$$

where the factors of 3 and 6 correspond to the number of possible orderings of three households of two and three distinct sizes respectively. Once a bubbled composition distribution is defined, we can then define a set of differential equations for the epidemic in this bubbled population exactly as we do for the standard, unbubbled population, with each risk class now corresponding to a different constituent household.

To simulate a temporary bubbling on day $t_b$, we first simulate the dynamics of the unbubbled population from day $t_0$ to day $t_b$ with some initial state distribution $\mathbf{H}_0$, using Eq 2, to obtain a final state distribution $\mathbf{H}(t_b)$. To simulate the formation of bubbles, we need to map the state distribution $\mathbf{H}(t_b)$ on the unbubbled population to one on the bubbled population, which we will denote $\mathbf{H}^b(t_b)$. For a two-household bubbling policy, this mapping is given by

$$H^b_{\mathbf{x}_1, \mathbf{x}_2,} = b_{(N_1, N_2)} \frac{H_{\mathbf{x}_1} H_{\mathbf{x}_2}}{p_{N_1} p_{N_2}}.$$

where we use the notation $N_i = N(\mathbf{x}_i)$ to denote the composition of a household with state $\mathbf{x}_i$. We can interpret this formula as saying that the proportion of bubbled households in a given

paired state $(\mathbf{x}_1, \mathbf{x}_2)$ is equal to the proportion of bubbled in the corresponding composition $(N_1, N_2)$ multiplied by the probability that the constituent households are in states $\mathbf{x}_1$ and $\mathbf{x}_2$, given they are of compositions $N_1$ and $N_2$. Using the same reasoning, for a three-household bubbling policy the mapping is given by

$$H^b_{\mathbf{x}_1,\mathbf{x}_2,\mathbf{x}_3} = b_{(N_1,N_2,N_3)} \prod_{i=1}^{3} \frac{H_{\mathbf{x}_i}}{p_{N_i}}.$$

We can then simulate a bubbling period beginning at time $t_b$ and ending at time $t_d$ (here $d$ stands for "dissolve"), by solving Eq 2 for the bubbled population with initial condition $\mathbf{H}^b(t_b)$. After the bubbling period, full lockdown measures are reimplemented, with mixing between distinct households forbidden. To simulate the infectious disease dynamics following the bubbling period, we need to map the bubbled system state at the end of the bubbling period, $\mathbf{H}^b(t_d)$ to the unbubbled model space. For the two-household bubble policy, this is done using the formula

$$H_{\mathbf{x}} = H^b_{\mathbf{x},\mathbf{x}} + \frac{1}{2} \sum_{\mathbf{y}\neq\mathbf{x}} H^b_{\mathbf{x},\mathbf{y}},$$

and for the three-household bubble policy this is done using the formula

$$H_{\mathbf{x}} = H^b_{\mathbf{x},\mathbf{x},\mathbf{x}} + \frac{1}{2} \sum_{\mathbf{y}\neq\mathbf{x}} H^b_{\mathbf{x},\mathbf{x},\mathbf{y}} + \frac{1}{2} \sum_{\mathbf{y}\neq\mathbf{x}} H^b_{\mathbf{x},\mathbf{y},\mathbf{x}} + \frac{1}{2} \sum_{\mathbf{y}\neq\mathbf{x}} H^b_{\mathbf{y},\mathbf{x},\mathbf{x}} + \frac{1}{3} \sum_{\mathbf{y}\neq\mathbf{x}} \sum_{\mathbf{z}\neq\mathbf{x}} H^b_{\mathbf{x},\mathbf{y},\mathbf{z}}.$$

Solving the unbubbled system forward in time allows us to simulate the dynamics of the epidemic following the bubbling period and thus make projections of its long term impact.

In what follows we model two basic interpretations of a short-term bubble policy. Under the first interpretation, bubbles of two or more households are formed every day for $T$ days, which each bubble dissolving and its constituent households joining new bubbles at the end of each day. Under the second, households are allowed to form bubbles of two or more households for $T$ days, after which the population returns to its previous "unbubbled" composition structure. The first roughly describes a situation in which individuals and their households visit a new household each day during the bubbling period, while the second describes a situation in which a group of households interact closely for an extended period, for instance with adult children staying at their parents' homes. From a network perspective, the first interpretation will allow giant components to form since we are rewiring on a daily basis, while the second involves replacing small household-level clusters with larger bubble-level clusters. Both interpretations offer a partial description of a population's actual behaviour over the Christmas period in a non-pandemic year and its behaviour during a bubbling period, both of which are likely to resemble a mixture of our model interpretations with more complex behaviours. We also model variations of both interpretations in which the initial bubbling period is followed by a second one-day period of bubbling, which captures the impact of relaxing isolation measures on New Year's Eve following a bubbling period over Christmas. In addition to these four potential scenarios, we also model the policy which was actually implemented, under which each household was allowed to meet with a single other household on December 25th, with no other exceptions to social contact restrictions. The five policies considered are thus as follows:

1. Each household forms a two-way bubble with one other household on December 25th only;

2. Each household forms a two-way bubble with one other household on each of December 25th and December 26th (so two households are bubbled with overall);

3. Each household forms a triangular bubble with two other households for two days (December 25th and 26th);

4. As in Policy 2, but each household also chooses a third household to bubble with on December 31st;

5. As in Policy 3, but each household also chooses a third household to bubble with on December 31st.

In our analysis we assume that every household joins a bubble, and that no bubbling other than that permitted by the policy takes place.

For each policy we initialised the self-consistent equations using the Euler-Lotka approach outlined in our methods section with prevalence and immunity both at 1%, consistent with estimates from the ONS COVID-19 dashboard in early December 2020 [62]. The between-household transmission parameter $\beta_{ext}$ is calibrated to a growth rate of $r = 2\%$, corresponding to SPI-M estimates from mid-December [63]. Setting all times relative to the start of 2020, so that at 12am January 1st $t = 0$, we solve the self-consistent equations for the unbubbled baseline population from $t = 335$ (12am on December 1st) to $t = 359$ (12am on December 25th). For each policy, we alternate between bubbled and unbubbled populations from $t = 359$ onwards according to the details of the policy, running all simulations up to $t = 396$ (12am on February 1st 2021). This allows us to observe the impact of each policy over the first few generations of post-bubbling cases. Because social contacts involving guests from outside the household are likely to differ from those with members of one specific household, we allow the density exponent $d$ to change between the baseline and bubbled populations. We carry out each policy analyses over a range of values of the two exponents to account for the uncertainty in the nature of these contacts. We also carry out this analysis for the baseline case where no relaxations are allowed.

The peak and cumulative prevalences over the December 1st to February 1st simulation period arising from each policy are plotted in Figs 5 and 6 respectively. In the absence of any temporary relaxations to NPIs (Figs 5a) and 6a)), we expect to see a peak prevalence of 1–2% and a cumulative prevalence of around 20% (including the initial 10% starting immunity) over the period December 1st to February 1st, depending on the level of density-dependence in transmission in single households. A single day of mixing between pairs of households results (policy 1, Figs 5b) and 6b)) in lower peak and cumulative prevalences than any of the other proposed relaxation policies. Although Policies 2 and 3 involve the same total mixing period, Policy 3 is associated with higher peak (Fig 5b) and 5c)) and cumulative prevalences(Fig 6b) and 6c)), suggesting that a prolonged period of mixing between multiple households is riskier than multiple shorter mixing periods. Fig 5e) and 5f), as well as Fig 6e) and 6f), demonstrate a small amplification in transmission arising from the extra day of mixing on January 31st. Overall these results suggest that short-term relaxations in mixing restrictions will have a small but non-negligible impact on the epidemic dynamics, with larger temporary bubbles and longer mixing periods associated with higher prevalences. The gradients of our plots suggest that the level of density dependence in single-household transmission has a greater impact on the long-term transmission dynamics than the level of density dependence in bubbled household transmission, and so our results should be fairly robust with respect to uncertainty over how interactions with visitors may differ from interactions with family members.

## Out of household isolation

In this section we analyse the impact of out-of-household isolation (OOHI) policies, under which detected cases are removed from their household until they have recovered from

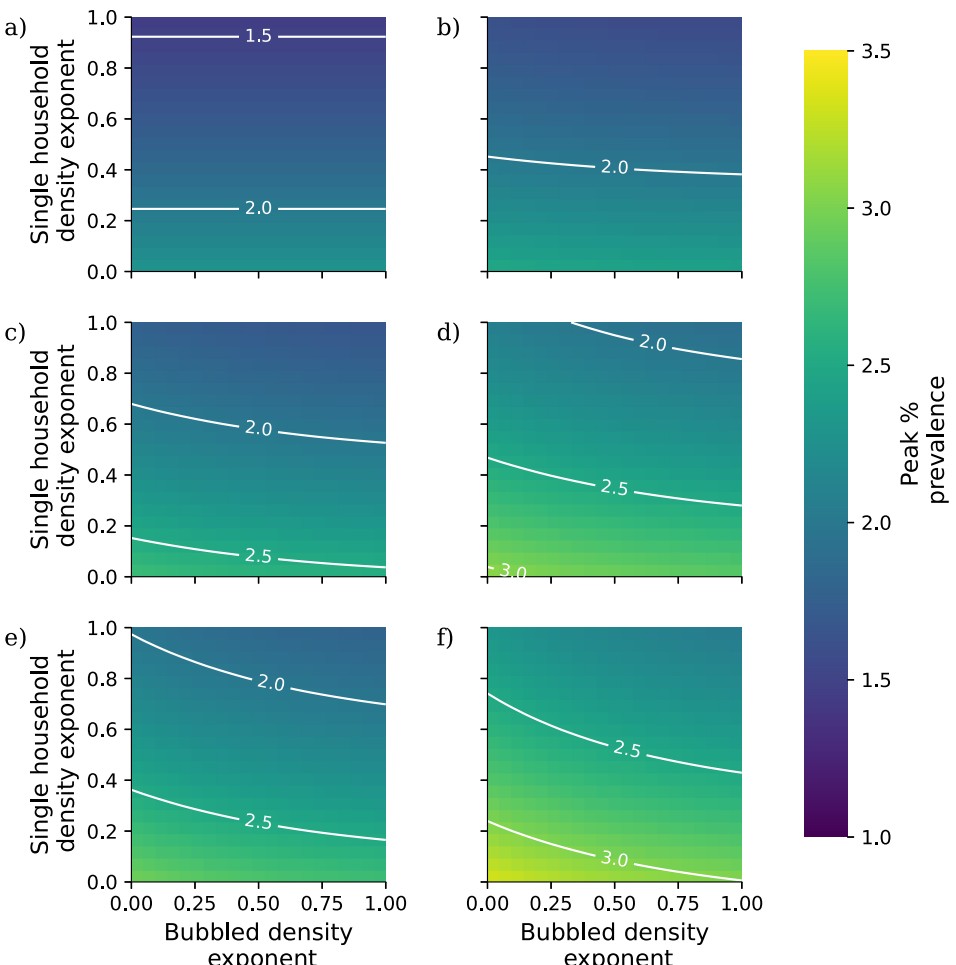

**Fig 5.** Peak prevalence between December 1st 2020 and February 1st 2021 as a function of density exponents in single households and in bubbled households under the following relaxation policies: a) no relaxation; b) two-household bubbles on December 25th; c) two-household bubbles on December 25th and again on December 26th; d) three-household bubbles from December 25th to December 26th; e) two-household bubbles on December 25th, again on December 26th, and again on December 31st; f) three-household bubbles from December 25th to December 26th and two-household bubbles on December 31st.

infection, with the intention of reducing the risk of infection to other members of the household. We specifically model the impact of OOHI when applied to the households of individuals who are identified as particularly vulnerable to infection, either due to age or chronic health conditions. The intention of such a policy would be to reduce the morbidity and mortality of infection by reducing the potential for transmission to those most at risk of infection; under more typical within-household isolation measures, isolation could potentially increase the risk of transmission to these individuals by increasing the time they spend in contact with infectious members of their own household. However, OOHI policies are inherently costly since they require governments to provide accommodation for isolating individuals, while also potentially placing strain on households by removing an individual who may be responsible for childcare or other aspects of household maintenance. Our goal here is therefore to assess both how effective an OOHI policy might be in reducing infection among clinically vulnerable

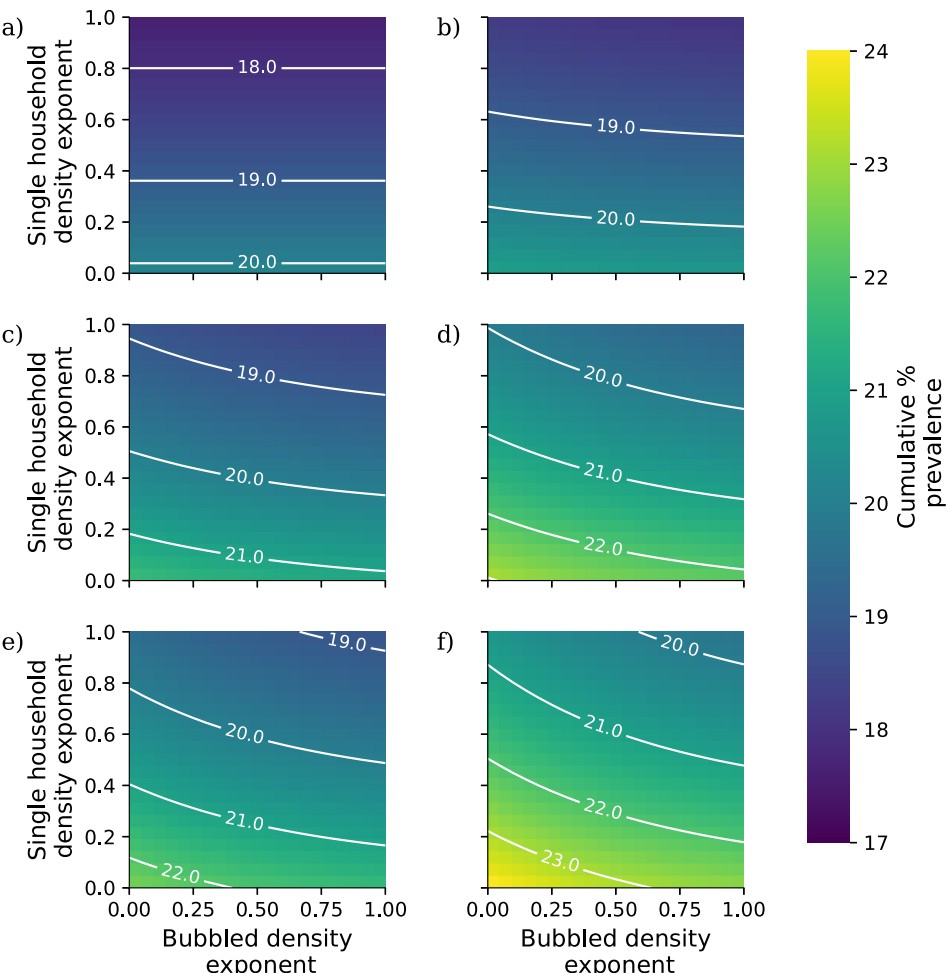

**Fig 6.** Cumulative prevalence between December 1st 2020 and February 1st 2021 as a function of density exponents in single households and in bubbled households under the following relaxation policies: a) no relaxation; b) two-household bubbles on December 25th; c) two-household bubbles on December 25th and again on December 26th; d) three-household bubbles from December 25th to December 26th; e) two-household bubbles on December 25th, again on December 26th, and again on December 31st; f) three-household bubbles from December 25th to December 26th and two-household bubbles on December 31st.

individuals, and also the number of individuals who will be required to isolate as part of such a strategy.

To model OOHI of infectious individuals, we expand upon our SEPIR compartmental structure to incorporate an isolation or Quarantine compartment $Q$, giving us a susceptible-exposed-prodromal-infectious-recovered-isolated (SEPIRQ) structure. Out-of-household isolation acts to temporarily remove a (detected) infectious individual from their household, so that the number of class $C_a$ individuals in a household is $N_a - Q_a$ rather than $N_a$. We divide the process of isolating into two stages. Under the simplest interpretation of the isolation process cases are detected at a rate $\delta$, so that the mean time to detection after being infected is $1/\delta$, and once detected an individual will isolate with probability $p_Q$. We assume that individuals in the Exposed, Prodromal, and Recovered compartments are all detected at the same rate, and that this rate is not class-specific. Because out-of-household isolation has the potential to be highly disruptive to family life and expensive to implement, we refine this interpretation so

that only individuals in certain classes are eligible to isolate, depending on the state $\mathbf{x}$ of the household. This allows us to do things like only isolate individuals who share a household with an individual in a particularly vulnerable class, and to stop adults from isolating if it means leaving children in their household without a caregiver. The isolation probability is thus defined to be a class-specific function of the household state $p_Q^a(\mathbf{x})$, such that $p_Q^a(\mathbf{x}) = p_Q$ if our policy is to isolate class $C_a$ individuals belonging to state $\mathbf{x}$ households and $p_Q^i(\mathbf{x}) = 0$ otherwise. Isolated individuals recover and return to their household at rate $\rho$, so that the mean isolation period is $1/\rho$ days. With these considerations in mind, the transition rates for the OOHI model are as follows:

$$(S_a, E_a, P_a, I_a, R_a, Q_a) \quad \rightarrow (S-1, E+1, P, I, R, Q_i) \text{ at rate}$$

$$S_a\left(\sum_b \beta_{\text{int}} k_{a,b}^{\text{int}} \frac{(\tau_b P_b + I_b)}{(N_b - Q_b)} + \Lambda_i(\mathbf{H}, t)\right) \tag{22}$$

$$(S_a, E_a, P_a, I_a, R_a, Q_a) \rightarrow (S, E-1, P+1, I, R, Q_i) \text{ at rate } \alpha_1 E \tag{23}$$

$$(S_a, E_a, P_a, I_a, R_a, Q_a) \rightarrow (S, E, P-1, I+1, R, Q_i) \text{ at rate } \alpha_2 P \tag{24}$$

$$(S_a, E_a, P_a, I_a, R_a, Q_a) \rightarrow (S, E, P, I-1, R+1, Q_i) \text{ at rate } \gamma I \tag{25}$$

$$(S_a, E_a, P_a, I_a, R_a, Q_a) \rightarrow (S, E-1, P, I, R, Q_i+1) \text{ at rate } \delta p_Q^i(\mathbf{x})E_i \tag{26}$$

$$(S_a, E_a, P_a, I_a, R_a, Q_a) \rightarrow (S, E, P-1, I, R, Q_i+1) \text{ at rate } \delta p_Q^i(\mathbf{x})P_i \tag{27}$$

$$(S_a, E_a, P_a, I_a, R_a, Q_a) \rightarrow (S, E, P, I-1, R, Q_i+1) \text{ at rate } \delta p_Q^i(\mathbf{x})I_i \tag{28}$$

$$(S_a, E_a, P_a, I_a, R_a, Q_a) \rightarrow (S, E, P, I, R+1, Q_i-1) \text{ at rate } \rho Q. \tag{29}$$

In our analysis we assume that any OOHI policy is implemented against the background of continuously applied within-household isolation policies. Under a within-household isolation policy, individuals isolated within their own home on testing positive for COVID-19. This reduces the intensity of between-household social contacts for detected infectious individuals. From the perspective of a susceptible individual, this will manifest as a reduction in the amount of between-household infection they are exposed to, which in turn manifests on the population level as a reduction in the epidemic growth rate. In our analysis of OOHI we calibrate the model to the estimated growth rate in the UK during August 2020, at which time self isolation policies were in place. Fitting the between-household transmission parameter $\beta_{\text{ext}}$ to this growth rate will implicitly account for the impact of within-household isolation. While our model could be adapted to perform a direct simulation of within-household isolation, with isolated individuals remaining within their own household but contributing nothing to the between-household force of infection, calibrating such an analysis to a growth rate estimated while within-household isolation policies were in place would effectively mean modelling the impact of these policies twice over.

Given uncertainty over how quickly cases can be detected and how likely individuals may be to isolate, we look at a range of scenarios with detection rates varying between 0.01 (detection takes on average ten days following infection) and 1 (detection takes on average one day following infection) and probability of isolation varying between 0 and 1. To estimate the percentage of each age group shielding, we used ONS shielding data by age cohort for England

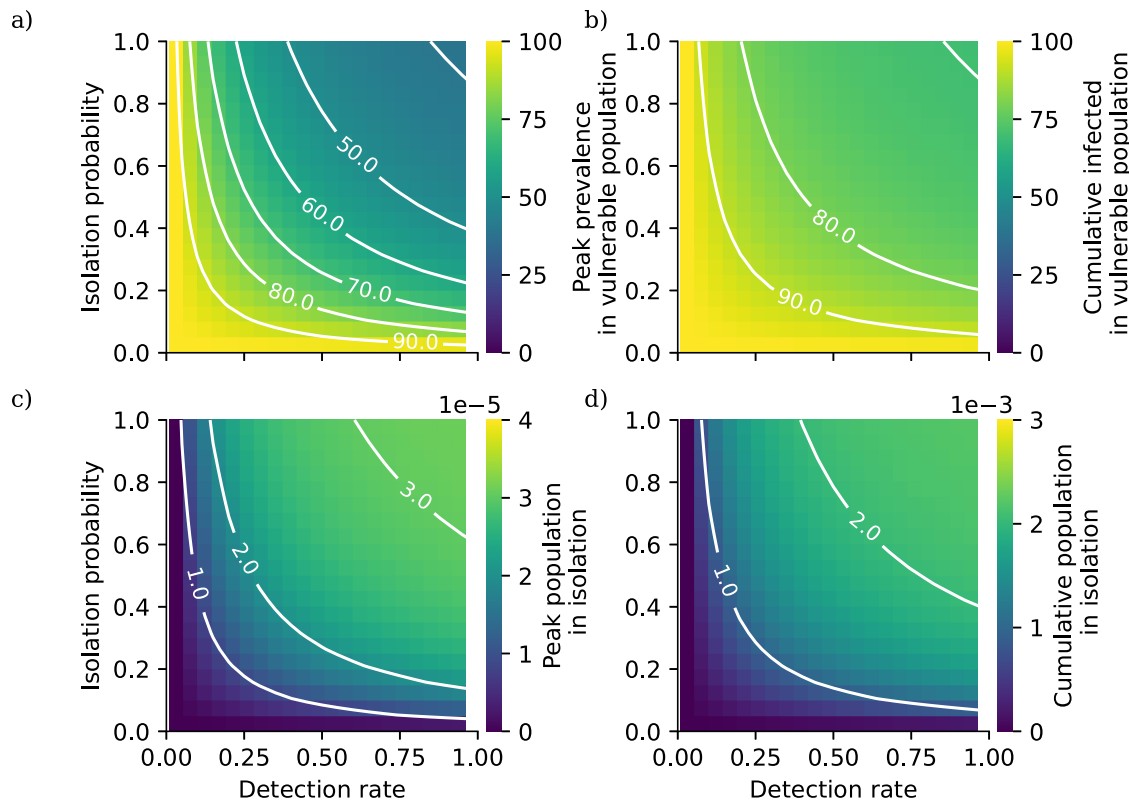

**Fig 7. Peak and b) cumulative prevalence among the clinically vulnerable population as a percentage of the baseline values when (OOHI) is not implemented, and c) peak and d) cumulative proportion of less clinically vulnerable adults who enter isolation during the simulation period as a function of detection rate of infectious cases and probability that a detected individual enters isolation.**

between July 9th and July 16th 2020 [52], divided by population estimates for England from mid 2019 to April 2020 [54]. Our between-household mixing rate is fit to a growth rate of negative 1%, corresponding to estimates from the UK around August 2020 [63], and our simulations are initialised with a prevalence of 1% and population immunity of 0.1%, consistent with estimates from the same period [62]. Our simulation period covers 100 days, starting from the introduction of the OOHI policy.

The results of this analysis are presented in Fig 7. In Fig 7a) and 7b), we plot the peak and cumulative prevalences in the clinically vulnerable population as proportions of their values when the OOHI policy is not implemented. Our simulations predict that these baseline values will be 0.009% and 0.244% respectively, reflecting the negative growth rate we have calibrated these simulations to. Fast detection and wide uptake of OOHI is projected achieve reductions in peak prevalence of up to 50%, although even with high levels of uptake and intensive enough surveillance to detect cases within a day of infection, OOHI is not predicted to reduce cumulative prevalence in this group below 70% of its baseline level. This suggests an OOHI policy's main benefit will be in reducing acute strain on health systems, while doing little to directly reduce total morbidity or mortality. In Fig 7c) and 7d) we plot the peak and cumulative proportions of the less vulnerable adult population who are projected to enter OOHI over the course of the simulation period. The population of the UK aged 20 and over as of 2018 is approximately 50 million [64], so that a peak on the order of $10-5$ corresponds to at least 500 people simultaneously in isolation, while a cumulative value on the order of $10^{-3}$ corresponds

to around 50 thousand people entering isolation over the simulation period. Comparing Fig 7a) and 7c) suggests that increasing detection speeds and isolation probabilities reduces peak prevalence in the vulnerable class faster than it increases the peak in number of people in isolation, in the sense that the proportion of cases averted (i.e. one minus the peak prevalence as a proportion of baseline) grows more quickly than the peak proportion of less vulnerable adults in isolation. Comparison of Fig 7b) and 7d) indicate a similar pattern in cumulative values. However, it is important to note that the probability that a detected case enters OOHI is unlikely to be directly controllable by public health authorities, and so these plots should be interpreted as suggesting a range of potential outcomes within a space of conceivable decision parameters, rather than the outcome of a public health strategy which can be directly optimised. In particular, high quarantine probabilities may be difficult to achieve in practice since individuals acting as carers to clinically vulnerable individuals may not be able to isolate due to those caring responsibilities.

## Discussion

In this study we have introduced a new approach to modelling infections in the household which is able to account for risk-stratified contact structures and response to infection. The self-consistent equations formulation of our infectious disease dynamics made it amenable to numerical analysis, unlike simulation-based approaches, allowing us to calibrate the within- and between-household transmission to household-level and population-level data respectively. The complex population structure of our model allows us to make predictions about the impact of a range of NPIs, illustrated by the four analyses we have presented. Our comparison of controls on transmission at the within- and between-household levels suggests that within-household measures are likely to have a limited impact due to the high proportion of non-household transmissions which characterise the spread of COVID-19. We also predict that allowing single-adult households to form support bubbles is not associated with a dramatic increase in transmission, offering a way to safely mitigate the impact of isolation during lockdown periods. Our analysis of temporary pauses in lockdown measures suggests that any lifting of measures is associated in an increase in cases, with triangular bubbles lasting two days resulting in more cases than repeated two-household contact events. Finally, our analysis of OOHI suggests that while external isolation can substantially reduce prevalence among clinically vulnerable individuals, the rapid spread of infection means that with high rates of detection we expect to see potentially infeasible numbers of individuals asked to isolate outside of their own home.

Our policy analyses made use of parallel computing to perform large numbers of simulations over a wide range of parameters. This is particularly important when modelling novel interventions whose impact is still uncertain. For instance, in our analysis of temporary pauses in NPIs, we performed simulations over a range of levels of density dependence. While we inferred the level of density dependence in within-household contacts from observational data, it is not obvious whether contacts with visitors to the household will exhibit the same behaviour, and our grid calculation approach allowed us to account for a spectrum of possibilities. In the analysis of OOHI, we explicitly account for uncertainty both in the rapidity of detection once a case develops, and the proportion of individuals who are able to isolate once identified. Estimates in this form can support intervention planning by pointing to levels of outreach or adherence which are necessary to drive infections below a specific threshold.

The modelling framework we have introduced here has several limitations, the main one being the large population ensemble limit which characterises the self-consistent equations framework. This large population assumption means our model is best suited to simulating

populations consisting of lots of statistically similar units, as has been the case throughout this study, where our population of interest consisted of all of the households in the UK. Our model will be less effective when the population of interest consists of a small number units, or units with greater variance in composition, as would be the case if we sought to simulate infectious disease dynamics in a network of connected institutions such as the wards in a hospital, the schools in an educational authority, or a country's prison network. While none of these example populations are households in the idiomatic sense, they all define a population of connected units with distinct levels of within-unit and between-unit mixing, which is the defining feature of a household-structured epidemic model [57]. A related limitation of our model is the scaling of system size with household size. Because our system of self-consistent equations requires one dimension for each possible system state, modelling large household sizes will come with a substantial computational cost. Similarly, populations with a finer age stratification or more complex divisions in terms of behavioural or clinical risk will require a model with more age classes than we have considered here, which itself increases the number of possible epidemiological states for a single household and thus the overall system size. We came up against these limits in our analysis of temporary bubble formation, where we ignored age stratification so that our model could deal with large bubbles consisting of two or three households. This potentially reduced the accuracy of our analysis, since during the Christmas period in the UK we could potentially see significant differences in contact behaviour between adults and children because school holidays around Christmas typically last for two weeks, as opposed to the three days of public holidays (December 25th and 26th as well as January 1st) which occur in this period. Extending the domain of applications of our model to populations with larger household sizes or finer risk structures will require us to address the computational limits of our model through further developments to our software implementation.

Another important factor to consider is the role of overdispersal in transmission, which has been observed as highly important for COVID-19 [65]. While households models naturally build some overdispersal into their next generation matrices due to stochastic effects within households [28], it may be possible to extend our model to consider additional heterogeneity in between-household mixing as considered by Ball *et al.* [66], although in this case we would expect significant additional dimensionality to the differential equation system describing the temporal dynamics.

An inherent requirement of our model is access to an accurate estimate of the household composition distribution of the population of interest. In the UK this is publicly available from the Office for National Statistics, although at the time our study was conducted the most recent estimates were from the 2011 census and will not reflect changes in the UK's demography which may have occurred since then, and this may reduce the accuracy of some of our model's projections. Calibrating our model requires access to estimates of both the epidemic growth rate and the SITP, which in turn rely on accurate reporting and surveillance of infection. In particular, forming an estimate of the SITP requires close surveillance of infections by household, which in the UK came as part of a large and unprecedented study [30]. This potentially limits our capacity to calibrate the within-household transmission dynamics to other national settings, particularly in cases where we expect social structures to differ dramatically from the UK.

In our policy analyses we have paid limited attention to questions of compliance to disease control regulations. Our analysis of support bubbles assumed that only eligible households would form bubbles, and that these bubbles would conform to the basic format laid out in the policy guidelines (one eligible household bubbling with one other household). Similarly, while our analysis of short-term bubbling over Christmas 2020 assumed full uptake of all mixing allowances, it also assumed that no households mixed with more households than they were

permitted to. These assumptions of perfect compliance to regulations are unlikely to account for the full range of responses to a given policy announcement. However, while our model structure inherently requires the number of possible responses to a given policy to be finite, there is no reason why these analyses could not be extended to include a richer variety of responses which do not necessarily conform to the letter of the law. As with most extensions to our model, a wider range of responses to policy has the potential to dramatically increase system size, and so a balance needs to be struck between capturing realistic behaviours and retaining a tractable model. We emphasise here that our parameter sweeps are intended to explore a range of possible responses to a given policy, rather than to illustrate the outputs of a control problem which can be tuned to a desired outcome. By simulating this range of responses, our model can provide upper and lower bounds on the potential impact of a policy and point to specific policies which may be more or less fruitful than others, as we saw in our analysis of controls within- and between-household transmission.

## Supporting information

**S1 Appendix.** Fig A in S1 Appendix. Current class structure of the software. Fig B in S1 Appendix. Convergence of model from initial conditions to exponential growth regime. Table A in S1 Appendix. Wall times for case study calculations.
(PDF)

## Acknowledgments

We thank Will Hart and Robin Thompson for their helpful comments on our choice of model parameters.

## Author Contributions

**Conceptualization:** Joe Hilton, Heather Riley, Lorenzo Pellis, Rabia Aziza, Samuel P. C. Brand, Ivy K. Kombe, John Ojal, Andrea Parisi, Matt J. Keeling, D. James Nokes, Robert Manson-Sawko, Thomas House.

**Data curation:** Joe Hilton, Heather Riley, Robert Manson-Sawko, Thomas House.

**Formal analysis:** Joe Hilton, Heather Riley, Lorenzo Pellis, Samuel P. C. Brand, Robert Manson-Sawko, Thomas House.

**Funding acquisition:** Joe Hilton, Matt J. Keeling, D. James Nokes, Robert Manson-Sawko, Thomas House.

**Investigation:** Joe Hilton, Robert Manson-Sawko, Thomas House.

**Methodology:** Joe Hilton, Heather Riley, Lorenzo Pellis, Rabia Aziza, Samuel P. C. Brand, Ivy K. Kombe, John Ojal, Andrea Parisi, Matt J. Keeling, D. James Nokes, Robert Manson-Sawko, Thomas House.

**Resources:** Robert Manson-Sawko.

**Software:** Joe Hilton, Heather Riley, Robert Manson-Sawko, Thomas House.

**Visualization:** Joe Hilton, Heather Riley, Robert Manson-Sawko.

**Writing – original draft:** Joe Hilton, Heather Riley, D. James Nokes, Robert Manson-Sawko, Thomas House.

**Writing – review & editing:** Joe Hilton, Thomas House.

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
