## [Decision Letter · Decision Letter 0]

12 Apr 2022

Dear Dr Hilton,

Thank you very much for submitting your manuscript "A computational framework for modelling infectious disease policy based on age and household structure with applications to the COVID-19 pandemic" for consideration at PLOS Computational Biology.

As with all papers reviewed by the journal, your manuscript was reviewed by members of the editorial board and by several independent reviewers. In light of the reviews (below this email), we would like to invite the resubmission of a significantly-revised version that takes into account the reviewers' comments.

We cannot make any decision about publication until we have seen the revised manuscript and your response to the reviewers' comments. Your revised manuscript is also likely to be sent to reviewers for further evaluation.

Sincerely,

Joseph T. Wu

Associate Editor

PLOS Computational Biology

Tom Britton

Deputy Editor

PLOS Computational Biology

Reviewer's Responses to Questions

**Comments to the Authors:**

Reviewer #1: Please see attached comments (submitted as a PDF).

Reviewer #2: Hilton and colleagues present a novel modelling framework that considers both within-household and between-household infectious disease transmission dynamics and is informed by a range of demographic and infection data. They demonstrate the utility and flexibility of their model via four policy analyses. The manuscript is well written and structured. The topic is of public health importance. The analyses are interesting and innovative. The methods seem logical and appropriate. However, I have made a number of comments and suggestions for the authors to consider:

General comments

1) Please provide more consistent sub-headings and descriptions of the four policy analyses throughout the manuscript so the reader can more easily comprehend your work.

For example, the introduction lists the following policy analyses:

* Control of within- and between-household mixing through NPIs

* Formation of support bubbles during lockdown periods

* Out-of-household isolation

* Temporary relaxation of NPIs during holiday periods

The methods section lists the following “complex” interventions:

*Long-term household bubbling - which is the “support bubbles”

*Short-term household bubbles - which is the temporary relaxation of NPIs during holiday periods

*Out-of-household isolation

The results and discussion section contains the following sub-sections:

*Impact of NPIs at the within- and between-household level

*Long-term social bubbles

*Short-term social bubbles

*Out of household isolation

The description of the first “policy analysis” is particularly confusing since all of the interventions explored in the study are NPIs. I suggest coming up with a more specific title for this analysis.

2) I’m not (yet) convinced of the utility of the “between-household” and “within-household” transmission controls analysis. It’s not clear to me how specific policies can be placed into these two categories. I wonder if the analysis would be more useful for policymakers if it were re-framed as “case targeted” and “community-wide” controls, with the careful consideration of within and between household transmission dynamics that your modelling framework allows. Case isolation and contact tracing and quarantine would be examples of case targeted controls and workplace closures and lockdowns would be examples of community-wide controls. Furthermore, have you considered that some community wide NPIs such as lockdowns may decrease between household transmission and increase within household transmission?

And depending on what you consider a “within-“ versus “between-household” control, I’m not sure how helpful the following conclusion is “...suggests that within-household measures likely to have a limited impact due to the high proportion of non-household transmissions which characterise the spread of COVID-19” - what specific measures are therefore likely to have a limited impact?

3) Please add further reflection of other relevant studies in the introduction and/or discussion. For example, the approach of other model-based NPI policy analyses.

4) Please add a discussion of the study limitations.

5) Please consider including a schematic overview at the start of the methods section showing the links between the model components, data inputs and policy analyses.

Specific comments

Methods

I’m not clear on the link(s) between the risk classes, within-household transmission and the parameterisation to estimates of SITP. How does susceptibility and infectiousness vary by risk class? Does the demographic composition of a household (and so risk class of individuals in a household) impact the SITP? How do these assumptions or estimates relate to other studies of age-specific susceptibility and transmissibility (e.g. Davies et al https://www.nature.com/articles/s41591-020-0962-9).

Line 183. What is the structure of the within-household social contact matrix? Is it informed by data?

Since you describe two ways in which the models are calibrated to observational studies (SITP and use of Euler-Lotka equation), please consider adding a sub-heading (around line 187) for the use of SITP estimates for parameterising within-household transmission (at the same heading level as the Euler-Lotka).

“Estimate the impact of NPIs on the growth in cases when population immunity is at low levels” (line 236), and “assume that repeated imports into the same household are rare in the early stages of the population-level outbreak” (line 248). Was this appropriate given the phase of the epidemic during the analysis period? When does this approach become inappropriate in terms of the epidemic phase? I think that this is an important discussion point.

Line 289 - “In Figure 1 we plot the early growth in cases” - by cases here, do you mean infections?

Lines 354-357. Please be more specific - what was the scaled back policy? And what is the more permissive approach that you explore. I realise that this later becomes more clear but worth stating from the outset of this section.

Line 360. “We expect that in the scenario we are modelling individuals are likely to mix intensively across risk classes within the household and will mostly not attend school or work, where ordinarily mixing is likely to be highly age-stratified.” Why? Because it’s over the Christmas period? What other NPIs were in place over this period?

Line 362. "This motivates our approximation that stratification by class is not required to capture transmission within a short-term bubble, and thereby keep the dimension of the system manageable" - I think that it would useful to include a brief discussion of if/how this might limit the applicability of your approach to other policy contexts in the discussion.

Near line 377 - “original NPIs are reimplemented…”. What are these?

Near line 380, first sentence of section 2.6.3 - “…giving us a SEPIR structure”. Should this be SEPIRQ?

And in the following sentence, “temporarily remove an individual” - is this clearer if revised to “…temporarily remove a (detected) infected individual…”?

Lines 382-385 - what is meant by the “within-household isolation policy” mentioned here - do the other members of the household quarantine because of the detection of an infection? Is that how between-household transmission is reduced? Wouldn’t the members of that household quarantine even under a policy of OOHI and so both policies reduce between household transmission but the OOHI also reduces within household transmission? Or do you assume individuals isolating within a household are able to reduce the probability of transmission to other household members? Please clarify the policies and thinking here.

And more broadly, are within-household isolation and contact quarantine accounted for in the policy analyses? Or were these policies not in place over the analysis period.

Results and discussion

Figure 3 caption - the caption says “between-household contact rates” - should these be transmission rates?

Line 467 - “the object of such a policy is chiefly to ameliorate the impact of NPIs…and to lessen the psychological impact of isolation on older people living alone” - please amend to isolation of people living alone. The impact of isolation applies to all single person households and you have modelled it as such.

Figures 6 and 7 - please be consistent in figure labels and caption descriptions. Currently plot labelled with single household and bubbled density exponents and caption describes pre-bubbling and during-bubbling density exponents.

Line 570-572 - “In practice, high quarantine probabilities…” - should this be isolation probabilities? Otherwise, I’m a little confused. And again lines 586-587 - “while external quarantine” - should this be isolation?

**Have the authors made all data and (if applicable) computational code underlying the findings in their manuscript fully available?**

Reviewer #1: Yes

Reviewer #2: Yes

PLOS authors have the option to publish the peer review history of their article (what does this mean?). If published, this will include your full peer review and any attached files.

Reviewer #1: No

Reviewer #2: No
---

## [Decision Letter · Decision Letter 1]

14 Jul 2022

Dear Dr Hilton,

We are pleased to inform you that your manuscript 'A computational framework for modelling infectious disease policy based on age and household structure with applications to the COVID-19 pandemic' has been provisionally accepted for publication in PLOS Computational Biology.

Best regards,

Joseph T. Wu

Associate Editor

PLOS Computational Biology

Tom Britton

Deputy Editor

PLOS Computational Biology

Reviewer's Responses to Questions

**Comments to the Authors:**

Reviewer #1: Thank you for your close attention to the review comments. The revised paper is much improved - in particular the structural changes make the paper much more readable, and the expanded discussion helps better establish the value of the model beyond the immediate applications.

Well done on a beautiful piece of work!

Only a few minor points that struck me as I was re-reading:

Abstract:

line 18: It may be personal preference, but I’d suggest a couple of commas:

“The widespread, and in many countries unprecedented, use of non-pharmaceutical interventions…”

line 24: “including” -> “that includes”

Author summary:

line 43: “a few” -> “four”

Introduction:

line 54: “NPIs” -> “non-pharmaceutical interventions (NPIs)” (realise it's in the abstract, but feel it should be spelt out in main body of paper too).

line 94: here you write “age- and household-structured” (which I would use); however, elsewhere you have written “age and household structured”; be consistent.

line 109: “study” -> “paper”

Equations are numbered inconsistently (some are, some aren’t).

Lines 486-517: this paragraph is verry long; this may be fine, but consider checking.

Reviewer #2: Thank you for addressing the reviewer comments so carefully. In particular, the re-structuring of the Methods and Results sections, and additional details on each of the policies analysed, has made the entire manuscript much easier to follow and my concerns related to the framing of the "between-household" and "within-household" transmission control analysis are now resolved.

I have a few very minor additional suggestions below:

Line 54. Suggest writing NPI out in full at first mention in intro.

Line 58. I would not consider social distancing a novel measure - e.g., think of school closures for influenza. It is also detailed in multiple countries’ pandemic planning documents prior to SARS-CoV-2 emergence. Perhaps be more specific? Are the social bubbles and shielding novel?

Line 88. age plays an important role in determining risk "of severe COVID-19 infection and death"??

Line 310 - "high prevalence of infection and minimal presence of variant strains" - is it worth also noting what transmission control policies were in place at this time?

Line 493 - I think that there is a typo - “…were advised NOT to leave their home”

Line 659 - one of the references appears to be missing and I’m not sure where the “CO:2020” belongs…

**Have the authors made all data and (if applicable) computational code underlying the findings in their manuscript fully available?**

Reviewer #1: Yes

Reviewer #2: Yes

PLOS authors have the option to publish the peer review history of their article (what does this mean?). If published, this will include your full peer review and any attached files.

Reviewer #1: No

Reviewer #2: No

---

## [Editor Report · Acceptance letter]

30 Aug 2022

PCOMPBIOL-D-22-00076R1 

A computational framework for modelling infectious disease policy based on age and household structure with applications to the COVID-19 pandemic

Dear Dr Hilton,

I am pleased to inform you that your manuscript has been formally accepted for publication in PLOS Computational Biology. Your manuscript is now with our production department and you will be notified of the publication date in due course.

With kind regards,

Zsofi Zombor
